# A Direct Second-Order Method for Solving Two-Player Zero-Sum Games

## Abstract

We introduce, to our knowledge, the first direct second-order method for computing Nash equilibria in two-player zero-sum games. To do so, we construct a Douglas-Rachford-style splitting formulation, which we then solve with a semi-smooth Newton (SSN) method. We show that our algorithm enjoys local superlinear convergence. In order to augment the fast local behavior of our SSN method with global efficiency guarantees, we develop a hybrid method that combines our SSN method with the state-of-the-art first-order method for game solving, Predictive Regret Matching$^+$ (PRM$^+$). Our hybrid algorithm leverages the global progress provided by PRM$^+$, while achieving a local superlinear convergence rate once it switches to SSN near a Nash equilibrium. Numerical experiments on matrix games demonstrate order-of-magnitude speedups over PRM$^+$ for high-precision solutions.

## 1 Introduction to BLSP

The Nash equilibrium is arguably the most foundational solution concept of game theory, representing a strategy profile in which no player can unilaterally deviate to improve their outcome. Approximating Nash equilibria of large-scale game models is the foundation of many human and superhuman-level AIs for games such as poker (Bowling et al., 2015; Brown & Sandholm, 2018; 2019), Stratego (Perolat et al., 2022), and Diplomacy (FAIR et al., 2022).

In the two-player zero-sum game setting, one player's gain is the other player's loss, so the game is completely described by a single payoff (utility) matrix $A \in \mathbb{R}^{n \times m}$, where entry $a_{ij}$ specifies the utility of the column player (and the negative utility of the row player) when the actions $i$ and $j$ are played. A (mixed-strategy) Nash equilibrium is then a pair of probability distributions $(x^*, y^*)$ over the rows and columns of $A$, such that $x^*$ and $y^*$ minimize and maximize their expected payoffs, respectively, given the strategy of the other player.

By von Neumann's minimax theorem, finding a Nash equilibrium in a two-player zero-sum game can be formulated as solving the bilinear saddle point problem (BLSP)

$$\min_{x \in X} \max_{y \in Y} x^\top A y \tag{1}$$

where $A \in \mathbb{R}^{n \times m}$ is the payoff matrix and $X \subseteq \mathbb{R}^n$, $Y \subseteq \mathbb{R}^m$ are convex, compact strategy sets for the min and max players.

In both normal-form and extensive-form games, $X$ and $Y$ are polytopes, and so the optimization problem in Equation (1) can be solved exactly by linear programming (Von Stengel, 1996). While this approach works in principle, it is impractical for large-scale games; even with state-of-the-art commercial solvers, the LP reformulation inflates the problem size and destroys exploitable structures in the payoff and constraint matrices, making exact solutions computationally prohibitive. Instead, first-order methods (FOMs) and regret minimization approaches have been widely used for large-scale game solving, where they have been highly successful in scaling to and solving large games arising from real-world applications. FOMs such as the *Extragradient Method* (Korpelevich, 1976) and *Mirror Prox* (Nemirovski, 2004) converge to a Nash equilibrium at a rate of $O(1/T)$, where $T$ is the number of iterations. Regret minimization approaches include *Regret Matching$^+$* (RM$^+$) and its predictive variant *Predictive Regret Matching$^+$* (PRM$^+$) (Zinkevich et al., 2007); these approaches use self-play to iteratively update each player's strategy proportional to

their positive cumulative regrets, and converge to a Nash equilibrium at a rate of $O(1/\sqrt{T})$, though their empirical performance is often much stronger. In fact, *Counterfactual Regret Minimization*$^+$ (CFR$^+$) (Tammelin, 2014) and its predictive variant PCFR$^+$ (Farina et al., 2021), which extend the RM$^+$ framework from normal-form to extensive-form games, are some of the fastest game-solving algorithms numerically, and CFR$^+$ served as the foundation for recent breakthroughs in superhuman poker AIs (Bowling et al., 2015; Brown & Sandholm, 2018; 2019).

While FOMs serve as the standard for computing approximate Nash equilibrium in two-player zero-sum games, their sublinear convergence guarantees make high-precision solutions expensive to compute. In convex minimization, high-precision solutions can be computed with second-order methods, which exploit curvature information for rapid local convergence. In two-player zero-sum game solving, however, there are no known "direct" methods for computing Nash equilibria using second-order methods. Classical second-order methods in optimization, such as interior-point algorithms, exploit curvature information through barrier formulations to achieve polynomial-time complexity for constrained problems. These methods can be applied by converting the BLSP into a linear program and applying the interior-point methods to that formulation. However, this throws away much of the structure of the BLSP and forces the use of generic methodology. Recently, there have been advances in applying second-order methods directly to saddle point problems, such as the $J$-symmetric quasi-Newton framework of Asl et al. (2024), but these advances are restricted to the unconstrained setting. In contrast, our method offers a direct means of incorporating curvature information into bilinear saddle point problems, without resorting to barrier reformulations or unconstrained analogues, while simultaneously leveraging the strengths of the state-of-the-art FOMs in game solving. As a first step to leverage the power of the curvature information inherent in the strategy spaces, we present a hybrid method combining a state-of-the-art FOM with a semi-smooth Newton method applied to an operator stemming from the Douglas-Rachford splitting algorithm (DRS). Our main results are as follows.

**Principled Use of Curvature Information.** To the best of our knowledge, we develop the first direct second-order method for solving bilinear saddle point problems, based on a Douglas–Rachford splitting formulation solved via a regularized semi-smooth Newton method. In contrast to other saddle point residual operators, our residual operator derived from DRS is monotone and piecewise affine, making it particularly well-suited for SSN methods. We establish that the regularized semi-smooth Newton method applied to our residual operator achieves local quadratic convergence. Our analysis replaces the classical BD-regularity assumption (Facchinei & Pang, 2003) with a local error bound condition and a stability property implied by monotonicity and Lipschitz continuity. In doing so, we extend prior metric-subregularity results for matrix games (Tseng, 1995; Gilpin et al., 2012; Wei et al., 2021) to our residual operator using tools from convex analysis.

**First and Second-order Hybrid Method.** While our second-order method has local superlinear convergence, it may be slow initially. To ameliorate this fact, we develop a principled approach for switching between iterates generated by a first-order method, which generally lie in the feasible set, and iterates for our SSN method, which generally move outside the feasible set. By systematically leveraging the progress of PRM$^+$ to warm-start our regularized semi-smooth Newton method, our approach provides the first framework combining first-order progress with second-order guarantees, ensuring efficient global progress together with local superlinear convergence as our method transitions to SSN steps. A key component of our design is a lifting framework that allows for an effective transition: given a strong FOM iterate, the corresponding lifted SSN iterate has a small residual norm, placing it squarely within the regime of local superlinear convergence.

**Numerical Experiments.** We conduct extensive numerical experiments on a diverse set of matrix games. For finding low-precision regimes, our hybrid method can be slower than pure FOMs due to a preprocessing inversion step and the overhead of solving Newton systems. However, once the SSN iterates enter a neighborhood of a Nash equilibrium in the lifted space, our hybrid method effectively exploits second-order information, leading to significant speedup over state-of-the-art FOMs. The results validate both the theoretical guarantees and the practical efficiency of our approach for high-precision equilibrium computation.

## 2 PRELIMINARIES AND TECHNICAL BACKGROUND

We provide the formal definitions for two-player zero-sum games considered in this paper, and the schemes of operator splitting methods and semi-smooth Newton methods.

### 2.1 TWO-PLAYER ZERO-SUM GAMES

We denote by $\mathcal{S} = \Delta^n \times \Delta^m$ the set of feasible strategy profiles, where $\Delta^n$ and $\Delta^m$ are the simplices representing the mixed strategies of the two players in an $n \times m$ zero-sum matrix game with payoff matrix $A$. A generic strategy profile will be written as $z := (x, y)$, with $x \in \Delta^n$ representing the minimizer's strategy and $y \in \Delta^m$ the maximizer's strategy. Within this framework, the duality gap serves as a natural measure of how far a strategy profile is from the set of Nash equilibria: it quantifies the incentive for either player to unilaterally deviate from their current strategies.

**Definition 1.** *For a strategy profile $\hat{z} = (\hat{x}, \hat{y})$, the duality gap is defined as*

$$\text{gap}(\hat{z}) = \max_{y \in \Delta^m} \hat{x}^\top A y - \min_{x \in \Delta^n} x^\top A \hat{y}. \tag{2}$$

*By definition, $\hat{z}$ is a Nash equilibrium if and only if $\text{gap}(\hat{z}) = 0$.*

The maximizer evaluates the highest payoff achievable by deviating from $\hat{y}$, while the minimizer evaluates the lowest payoff achievable by deviating from $\hat{x}$, demonstrating that $\text{gap}(\hat{z}) \geq 0$ always holds. In practice, iterative algorithms for computing Nash equilibria seldom converge exactly to an equilibrium, but rather produce approximate solutions. This motivates the notion of an $\epsilon$-Nash equilibrium, where the incentives to deviate are bounded by a small tolerance $\epsilon > 0$.

**Definition 2.** *A strategy profile $\hat{z} = (\hat{x}, \hat{y})$ is an $\epsilon$-Nash equilibrium if the following condition holds:*

$$\max_{y \in \Delta^m} \hat{x}^\top A y - \epsilon \leq \hat{x}^\top A \hat{y} \leq \min_{x \in \Delta^n} x^\top A \hat{y} + \epsilon.$$

**Remark 1.** *The concept of an $\epsilon$-Nash equilibrium can be equivalently expressed in terms of the duality gap (Orabona, 2019). Specifically, if $\hat{z}$ is an $\epsilon$-Nash equilibrium, then its duality gap satisfies $\text{gap}(\hat{z}) \leq 2\epsilon$. Conversely, if a strategy profile $\hat{z}$ achieves a duality gap below $\epsilon$, then $\hat{z}$ itself constitutes an $\epsilon$-Nash equilibrium. This tight connection establishes the duality gap as a natural and widely used performance metric for analyzing iterative algorithms.*

### 2.2 OPERATOR SPLITTING METHODS

As an alternative to the duality gap, Nash equilibria can also be characterized through residual operators. A natural choice is the classical saddle-point normal map $z \mapsto z - \Pi_{\mathcal{S}} \left( z - \eta \begin{bmatrix} 0 & A \\ -A^\top & 0 \end{bmatrix} z \right)$, but this map is not monotone in the two-player zero-sum game setting and therefore lacks a desired structural property for SSN methods. Instead, we base our construction on the optimality condition for two-player zero-sum games: a strategy profile $\hat{z} = (\hat{x}, \hat{y}) \in \mathcal{S}$ is a Nash equilibrium if and only if $0 \in F(\hat{z}) + N_{\mathcal{S}}(\hat{z})$ where

$$F(\hat{z}) = \begin{bmatrix} 0 & A \\ -A^\top & 0 \end{bmatrix} \hat{z}, \quad N_{\mathcal{S}}(\hat{z}) = \{v \mid v^\top (z - \hat{z}) \leq 0 \quad \forall z \in \mathcal{S}\}.$$

Thus, finding a Nash equilibrium reduces to finding a root of the sum of two structured operators. From standard convex analysis, both $F(\cdot)$ and $N_{\mathcal{S}}(\cdot)$ are maximally monotone, which guarantees that their resolvents $J_F^\gamma(\cdot) = (I + \gamma F)^{-1}(\cdot)$ and $J_{N_{\mathcal{S}}}^\gamma = (I + \gamma N_{\mathcal{S}})^{-1}(\cdot)$ are single-valued and firmly nonexpansive. This fact underpins the use of operator-splitting methods in solving two-player zero-sum games; for an overview of such methods, see the monographs Rockafellar & Wets (1998); Bauschke & Combettes (2017); Ryu & Yin (2022).

In particular, we focus on the Douglas-Rachford splitting (DRS) method (Lions & Mercier, 1979; Eckstein & Bertsekas, 1992), which applies to sums of two maximal monotone operators. With step size $\gamma > 0$, the DRS operator applied to our setting is given by

$$T_{\text{DRS}}^\gamma = \text{Id} - J_{N_{\mathcal{S}}}^\gamma + J_F^\gamma \circ (2 J_{N_{\mathcal{S}}}^\gamma - \text{Id}), \tag{3}$$

where Id denotes the identity operator. Because $F(\cdot)$ is linear and $N_{\mathcal{S}}(\cdot)$ is a normal cone operator, their resolvents simplify to

$$J_F^\gamma(z) = \begin{bmatrix} I & \gamma A \\ -\gamma A^\top & I \end{bmatrix}^{-1} z, \quad J_{N_{\mathcal{S}}}^\gamma(z) = \Pi_{\mathcal{S}}(z).$$

Thus, $J_F^\gamma$ is a linear operator and $J_{N_{\mathcal{S}}}^\gamma$ is simply the projection onto $\mathcal{S}$. As a consequence, one DRS iteration takes the following form of

$$z_{k+1} = T_{\text{DRS}}^\gamma(z_k) = z_k - \Pi_{\mathcal{S}}(z_k) + \begin{bmatrix} I & \gamma A \\ -\gamma A^\top & I \end{bmatrix}^{-1} (2\Pi_{\mathcal{S}}(z_k) - z_k). \tag{4}$$

We define the residual operator $R_{\text{DRS}}^\gamma := \text{Id} - T_{\text{DRS}}^\gamma$, noting that $R_{\text{DRS}}^\gamma(\hat{z}) = 0$ implies $\Pi_{\mathcal{S}}(\hat{z})$ is a Nash equilibrium (see Section 3.1). The following lemma summarizes one of its key properties.

**Lemma 1.** *The operator $R_{\text{DRS}}^\gamma(\cdot)$ is*

   *(i) monotone:* $(z_1 - z_2)^\top (R_{\text{DRS}}^\gamma(z_1) - R_{\text{DRS}}^\gamma(z_2)) \geq 0$

   *(ii) 1-Lipschitz:* $\|R_{\text{DRS}}^\gamma(z_1) - R_{\text{DRS}}^\gamma(z_2)\| \leq \|z_1 - z_2\|$

*for all $z_1, z_2$.*

Our approach is to solve the root-finding problem for this residual operator using semi-smooth Newton (SSN) methods (Qi & Sun, 1993; Pang & Qi, 1993; Martínez & Qi, 1995). This shares a similar spirit with prior SSN applications in composite convex optimization (Xiao et al., 2018) and kernel-based optimal transport (Lin et al., 2024).

However, the unique structure of two-player zero-sum games requires new algorithmic strategies, including nontrivial switching between SSN and PRM$^+$, making the practical implementation considerably more delicate.

## 2.3 SEMI-SMOOTH NEWTON METHODS

Semi-smooth Newton (SSN) methods generalize the classical Newton method to nonsmooth equations by replacing the Jacobian with a generalized Jacobian. Since $R_{\text{DRS}}^\gamma(\cdot)$ is Lipschitz continuous, Rademacher's theorem implies that it is differentiable almost everywhere. This motivates the use of the notion of a generalized Jacobian (Clarke, 1990).

**Definition 3.** *Let $F$ be a Lipschitz mapping and $D_F$ the set of differentiable points of $F$. The B-subdifferential at $z$ is $\partial_B F(z) = \{\lim_{k \to +\infty} \nabla F(z^k) \mid z^k \in D_F, z^k \to z\}$ and the generalized Jacobian is defined as $\partial F(z) = \text{conv}(\partial_B F(z))$, where conv denotes the convex hull.*

A second key notion is semi-smoothness, introduced by Mifflin (1977) for real-valued functions and extended to vector-valued mappings by Qi & Sun (1993).

**Definition 4.** *Let $F$ be a Lipschitz mapping. Then $F$ is (strongly) semi-smooth at $z$ if (i) $F$ is directionally differentiable at $z$, and (ii) for any $G \in \partial F(z + \Delta z)$, as $\Delta z \to 0$,*

$$\begin{aligned} \text{(semismooth)} \quad & \frac{\|F(z+\Delta z) - F(z) - G\Delta z\|}{\|\Delta z\|} \to 0, \\ \text{(strongly semismooth)} \quad & \frac{\|F(z+\Delta z) - F(z) - G\Delta z\|}{\|\Delta z\|^2} \leq C \end{aligned}$$

*for some constant $C > 0$.*

The following lemma shows that $R_{\text{DRS}}^\gamma(\cdot)$ is strongly semi-smooth and guarantees that the SSN method is suitable to solve $R_{\text{DRS}}^\gamma(z) = 0$.

**Lemma 2.** *The operator $R_{\text{DRS}}^\gamma(\cdot)$ is strongly semi-smooth.*

This property justifies the use of SSN to solve $R_{\text{DRS}}^\gamma(z) = 0$. In particular, the regularized SSN method proceeds as follows: given $z_k$, compute $z_{k+1} = z_k + \Delta z_k$, where $\Delta z_k$ solves

$$(G_k + \mu_k I)\Delta z_k = -r_k, \tag{5}$$

with $G_k \in \partial R_{\mathrm{DRS}}^\gamma(z_k)$, $r_k = R_{\mathrm{DRS}}^\gamma(z_k)$, and $I$ the identity matrix. The regularization parameter is chosen adaptively as $\mu_k = \theta_k |r_k|$ for $\theta_k > 0$. This stabilizes the method in practice: as $r_k \to 0$, the algorithm gradually exploits curvature information from $G_k$ while maintaining invertibility of $G_k + \mu_k I$.

If $R$ is continuously differentiable and $\theta_k = 0$, this reduces to the Newton method, which exhibits local quadratic convergence. Although regularized SSN methods may diverge in general (Kummer, 1988), local superlinear convergence is guaranteed under strong semi-smoothness and local error bound conditions (Zhou & Toh, 2005).

## 3 MAIN RESULTS

In this section, we first establish a formal connection between Nash equilibria and lifted fixed points, which serves as the theoretical foundation of our approach. Second, building on this insight, we introduce a new game-solving algorithm that integrates PRM$^+$ with a regularized semi-smooth Newton method. Finally, we provide a convergence analysis of the resulting procedure.

### 3.1 LINKING NASH EQUILIBRIA TO LIFTED FIXED POINTS

While Nash equilibria are not themselves fixed points of $T_{\mathrm{DRS}}^\gamma$, they can be linked through a simple lifting, as the following theorem shows.

**Theorem 1.** *The following relationships between fixed points of $T_{\mathrm{DRS}}^\gamma$ and Nash equilibria hold:*

*(i) If $z \in \mathbb{R}^{m+n}$ is a fixed point of $T_{\mathrm{DRS}}^\gamma$, then $\hat{z} = \Pi_{\mathcal{S}}(z)$ is a Nash equilibrium.*

*(ii) If $\hat{z} \in \mathcal{S}$ is a Nash equilibrium, then $z = \hat{z} - \gamma F(\hat{z})$ is a fixed point of $T_{\mathrm{DRS}}^\gamma$.*

**Remark 2.** *A Nash equilibrium is not necessarily a fixed point of $T_{\mathrm{DRS}}^\gamma$. Indeed, since $\Pi_{\mathcal{S}}(\hat{z}) = \hat{z}$, we have*

$$T_{\mathrm{DRS}}^\gamma(\hat{z}) = \hat{z} - \Pi_{\mathcal{S}}(\hat{z}) + \begin{bmatrix} I & \gamma A \\ -\gamma A^\top & I \end{bmatrix}^{-1} (2\Pi_{\mathcal{S}}(\hat{z}) - \hat{z}) = \begin{bmatrix} I & \gamma A \\ -\gamma A^\top & I \end{bmatrix}^{-1} \hat{z}.$$

*Thus, $\hat{z}$ is a fixed point of $T_{\mathrm{DRS}}^\gamma$ if and only if $A = 0$, which does not hold in general. The role of the lifted point $z = \hat{z} - \gamma F(\hat{z})$ is to displace $\hat{z}$ along its normal cone direction, which ensures the fixed-point relation.*

To implement the SSN method, we require an efficient way to obtain one element in $\partial R_{\mathrm{DRS}}^\gamma$. By definition, we have

$$R_{\mathrm{DRS}}^\gamma = J_{N_{\mathcal{S}}}^\gamma - J_F^\gamma \circ (2J_{N_{\mathcal{S}}}^\gamma - \mathrm{Id}),$$

where

$$J_F^\gamma(z) = \begin{bmatrix} I & \gamma A \\ -\gamma A^\top & I \end{bmatrix}^{-1} z, \quad J_{N_{\mathcal{S}}}^\gamma(z) = \Pi_{\mathcal{S}}(z).$$

The generalized Jacobian of $R_{\mathrm{DRS}}^\gamma(\cdot)$ at $z$ is given by

$$\partial R_{\mathrm{DRS}}^\gamma(z) = \partial J_{N_{\mathcal{S}}}^\gamma(z) - (\partial J_F^\gamma) \circ (2\partial J_{N_{\mathcal{S}}}^\gamma(z) - z) \cdot (2\partial J_{N_{\mathcal{S}}}^\gamma(z) - I),$$

where

$$\partial J_F^\gamma(z) = \begin{bmatrix} I & \gamma A \\ -\gamma A^\top & I \end{bmatrix}^{-1}, \quad \partial J_{N_{\mathcal{S}}} = \partial \Pi_{\mathcal{S}}(z) = \left\{ \begin{bmatrix} G_x & 0 \\ 0 & G_y \end{bmatrix} : G_x \in \partial \Pi_{\Delta^n}(x), G_y \in \partial \Pi_{\Delta^m}(y) \right\}.$$

Thus, finding an element in $\partial R_{\mathrm{DRS}}^\gamma$ reduces to finding an element of the generalized Jacobian of the simplex projection operator.

**Theorem 2.** *Let $p \in \mathbb{R}^d$ and let $x = \Pi_{\Delta^d}(p)$ denote the projection of $p$ onto the simplex $\Delta^d$. Define the active set $\mathcal{A} := \{i \in [d] : x_i > 0\}$. Then*

$$G = \mathrm{diag}(a) - \frac{1}{\|a\|_1} aa^\top,$$

*where $a \in \mathbb{R}^d$ is the indicator vector of the active set $\mathcal{A}$, satisfies $G \in \partial \Pi_{\Delta^d}(p)$.*

Putting these pieces together yields

$$\begin{bmatrix} G_x & 0 \\ 0 & G_y \end{bmatrix} - \begin{bmatrix} I & \gamma A \\ -\gamma A^\top & I \end{bmatrix}^{-1} \left( 2 \begin{bmatrix} G_x & 0 \\ 0 & G_y \end{bmatrix} - I \right) \in \partial R_{\mathrm{DRS}}^\gamma(z). \tag{6}$$

Theorem 1 establishes that projecting a fixed point of $T_{\mathrm{DRS}}^\gamma$ onto $\mathcal{S}$ yields a Nash equilibrium, and conversely, every Nash equilibrium induces a lifted fixed point. These results demonstrate a one-to-one correspondence between exact fixed points and exact Nash equilibria.

In practice, however, algorithms rarely reach exact solutions; instead, they operate with approximate fixed points and $\epsilon$-Nash equilibria. We therefore extend this correspondence to the approximate setting. In particular, we show how the residual norm of $R_{\mathrm{DRS}}^\gamma(\cdot)$ can be related to the duality gap. This link guarantees that progress made by a first-order method (which decreases the duality gap) translates into progress for the semi-smooth Newton method (which works to decrease the residual norm), and vice versa. As such, the connection enables a principled transition between the two regimes within our proposed algorithmic framework.

To formalize this relationship, we require the following condition measure of the payoff matrix $A$, which is used to control the Euclidean distance to the set of equilibria in terms of the duality gap. Being able to relate the duality gap to the iterate distance serves as an essential tool in our analysis.

**Definition 5** (Condition Measure (Gilpin et al., 2012))**.** *The condition measure of a matrix $A$, denoted $\delta(A)$, is defined as*

$$\delta(A) = \sup \left\{ \delta : \mathrm{dist}(\hat{z}, \hat{Z}^\star) \leq \frac{\mathrm{gap}(\hat{z})}{\delta} \text{ for all } \hat{z} \in \mathcal{S} \right\}.$$

**Remark 3.** *In addition, Gilpin et al. (2012, Lemma 4) guarantees that there exists $\delta > 0$ such that*

$$\mathrm{dist}(\hat{z}, \hat{Z}^\star) \leq \frac{\mathrm{gap}(\hat{z})}{\delta} \tag{7}$$

*for all $\hat{z} \in \mathcal{S}$. Thus, we have $\delta(A) < +\infty$ for any two-player zero-sum game.*

**Theorem 3.** *Let $Z^*$ denote the set of zeros of $R_{\mathrm{DRS}}^\gamma(\cdot)$, let $\hat{Z}^*$ denote the set of Nash equilibria of $\mathcal{S}$, and let $\hat{z} \in \mathcal{S}$. The following relationships hold:*

*(i) If $\|R_{\mathrm{DRS}}^\gamma(z)\| \geq \tau \mathrm{dist}(z, Z^\star)$ for some constant $\tau > 0$, we have*

$$\mathrm{gap}(\Pi_\mathcal{S}(z)) \leq \frac{\sqrt{2}\|A\| \cdot \|R_{\mathrm{DRS}}^\gamma(z)\|}{\tau}.$$

*(ii) For the lifted point $z = \hat{z} - \gamma R_{\mathrm{DRS}}^\gamma(\hat{z})$, we have*

$$\|R_{\mathrm{DRS}}^\gamma(z)\| \leq \frac{(1 + \gamma\|A\|)\mathrm{gap}(\hat{z})}{\delta(A)}.$$

## 3.2 Algorithmic Framework

We now outline the general framework of our algorithm. The core idea is to warm-start our Douglas-Rachford-based regularized SSN method (DRSSN) using iterates generated by a phase of exclusively PRM⁺ steps. The PRM⁺ phase reduces the duality gap to bring iterates sufficiently close to a Nash equilibrium, thereby avoiding the instability that can arise when DRSSN is initialized by lifting an iterate arbitrarily far from a Nash equilibrium, starting a regime where curvature information is not yet informative. Moreover, the steps in the PRM⁺ phase can be used to update the damping parameter for DRSSN. After the duality gap drops below a prescribed threshold, our algorithm makes a one-time switch from PRM⁺ to DRSSN, after which only Newton steps are taken. This design ensures global efficiency from the first-order phase and rapid local convergence from the second-order phase.

The DRSSN subroutine implements our regularized semi-smooth Newton method. Each iteration involves solving a linear system to compute the Newton step $\Delta z_k$, followed by a damping parameter update that depends on changes in the residual norm $\|R_{\mathrm{DRS}}^\gamma\|$. While the duality gap is the standard measure of proximity to a Nash equilibrium and is natural for first-order methods that remain within

---

**Algorithm 1** Hybrid SSN method

---

1: **Input**: $p_0 = z_0 \in \Delta^n \times \Delta^m$, $\theta_0 > 0$, $\gamma > 0$
2: Calculate resolvent for DRSSN method
3: **for** $k = 0, 1, \ldots$ **do**
4:      Update $p_{k+1} = p_k$ using one-step of PRM$^+$.
5:      **if** ready to update $\theta_k$ **then**
6:          Update $\theta_k$ adaptively.
7:      **if** ready to switch to DRSSN method **then**
8:          Set $z_k = p_k - \gamma F(p_k)$.
9:          Run DRSSN$(z_k, \theta_k)$.

---

**Algorithm 2** DRSSN$(z, \lambda)$

---

1: **Input**: $z_0 = z \in \mathbb{R}^{n+m}$, $\lambda_0 = \lambda > 0$, $\ell = 1.5$
2: **for** $k$ in $0, 1, \ldots,$ **do**
3:      **while** $\lambda_k \leq 10^9$ **do**
4:          Select $G_k \in \partial R^\gamma_{\mathrm{DRS}}(z_k)$.
5:          Use $G_k$ and solve the linear system in Equation (5) for $\Delta z_k$.
6:          Define $z' := z_k + \Delta z_k$.
7:          **if** $\|R^\gamma_{\mathrm{DRS}}(z')\| < \|R^\gamma_{\mathrm{DRS}}(z_k)\|$ **then**
8:              Set $z_{k+1} = z'$.
9:              Update $\lambda_{k+1} = \lambda_k \cdot \ell$.
10:             **break**
11:          **else**
12:              Update $\lambda_{k+1} = \lambda_k/\ell$.
13:      Update $\lambda_{k+1}$ using the adaptive scheme.
14: **if** gap$(\Pi_{\mathcal{S}}(z_{k+1})$ is under desired gap **then**
15:      **return** $\Pi_{\mathcal{S}}(z_{k+1})$.

---

the feasible strategy spaces, it is less effective for guiding second-order methods. In particular, the SSN iterates may leave the feasible region, and relying solely on the projected duality gap may fail to capture true progress. Instead, effective performance of our SSN method hinges on proper tuning of the damping parameter to control the residual norm, which in turn governs its convergence.

This distinction is crucial in our hybrid algorithm. The first-order phase reduces the duality gap to a chosen threshold, ensuring a stable lifted iterate for the second-order phase. In particular, if the duality gap of the PRM$^+$ iterates is sufficiently small, then by Theorem 3, the residual norm of the corresponding lifted point is appropriately bounded, ensuring that it lies within the regime where our SSN theory guarantees local superlinear convergence. The second-order phase advances through Newton steps while adaptively adjusting the damping parameter based on the residual norm, thereby exploiting curvature information to achieve accelerated convergence. To this end, our SSN method employs two damping-update strategies: (i) a line-search regime, which adjusts the parameter according to whether the residual norm decreases, thus ensuring consistent global progress when iterates are far from equilibrium; and (ii) an adaptive regime, inspired by Lin et al. (2024), which modifies the parameter heuristically to stabilize performance when the line-search regime stalls. Further details on the algorithmic details and the adaptive scheme are provided in the appendix.

### 3.3 CONVERGENCE GUARANTEE

We now focus on the convergence guarantees of our hybrid algorithm. For ease of reference, we denote $\mathrm{dist}(z, Z^*) := \min_{z^* \in Z^*} \|z - z^*\|$.

The SSN steps of our hybrid algorithm are applied to the residual operator $R^\gamma_{\mathrm{DRS}}(\cdot)$, which enjoys several structural properties that enable a local superlinear convergence guarantee. In particular, by Lemma 1, $R^\gamma_{\mathrm{DRS}}(\cdot)$ is both monotone and Lipschitz continuous, so every choice $G \in \partial R^\gamma_{\mathrm{DRS}}(z)$ in an SSN step will yield a positive semidefinite matrix (Xiao et al., 2018, Lemma 3.5). Then, the below stability condition is immediately satisfied (Zhou & Toh, 2005, Lemma 3.1).

**Lemma 3.** *For $G \in \partial R_{\mathrm{DRS}}^{\gamma}(z)$ and any $\mu > 0$, $\|(G + \mu I)^{-1}\| \leq \frac{1}{\mu}$.*

Moreover, $R_{\mathrm{DRS}}^{\gamma}(\cdot)$ is piecewise affine. Consequently, it satisfies both strong semi-smoothness (Lemma 2) and a local error-bound condition described in Lemma 4.

**Lemma 4.** *For any $z$ in a local neighborhood around a point $z^* \in Z^*$,*

$$\|R_{\mathrm{DRS}}^{\gamma}(z)\| \geq \tau \operatorname{dist}(z, Z^*).$$

This local error-bound condition is also known as local metric subregularity. In the context of matrix games, both the duality gap operator (Gilpin et al., 2012) and the gradient operator $F(z)$ (Wei et al., 2021) have been shown to satisfy variants of this property. In our setting, the residual operator is piecewise affine and thus polyhedral, which by Robinson (2009) implies local metric subregularity. Thus, we establish a direct extension of prior results and show that the analysis applies more generally to any polyhedral mapping.

These conditions help enforce the local quadratic convergence of the Newton steps of our hybrid algorithm.

**Theorem 4.** *The regularized semi-smooth Newton method defined by the update rule in Equation (5) applied to $R_{\mathrm{DRS}}^{\gamma}(\cdot)$ exhibits local quadratic convergence. Concretely, for $k$ sufficiently large (so that the iterates are in a local region) and iterates $\{z_k\}_{k \in \mathbb{N}}$, we have*

$$\|R_{\mathrm{DRS}}^{\gamma}(z_{k+1})\| \leq C \|R_{\mathrm{DRS}}^{\gamma}(z_k)\|^2 \tag{8}$$

*where $C = \frac{1}{\tau^2} \left[ L_2 \left( 2 + \frac{L_2}{\theta \tau} \right)^2 + \theta L_1 \left( 2 + \frac{L_2}{\theta \tau} \right) \right]$.*

**Remark 4.** *Note that the above result does not rely on $\theta$ being fixed (see Appendix A.2). Consequently, the final bound in Equation (8) also holds in the case where each $\theta_k \in [\underline{\theta}, \overline{\theta}]$, i.e. bounded between fixed lower and upper bounds for $\theta$. This holds in many adaptive methods (Xiao et al., 2018; Lin et al., 2024), and ours, which vary $\theta$ at each time step.*

While the above analysis establishes the local quadratic convergence of our residual operator, the analysis applies more generally to any operator satisfying the stated conditions. Our analysis builds on the framework of Zhou & Toh (2005) and extends classical SSN theory by replacing the restrictive BD-regularity assumption (Xiao et al., 2018) with weaker and more broadly applicable conditions. We outline these conditions in detail below, which must be satisfied for any $z_1, z_2 \in \operatorname{dom}(F)$.

**Condition 1.** *$F$ is $L_1$-Lipschitz: for a constant $L_1 > 0$,*

$$\|F(z_1) - F(z_2)\| \leq L_1 \|z_1 - z_2\|.$$

**Condition 2.** *$F$ is strongly semi-smooth: for a constant $L_2 > 0$,*

$$\|F(z_1) - F(z_2) - G(z_1 - z_2)\| \leq L_2 \|z_1 - z_2\|^2$$

*for $G \in \partial F(z_1)$.*

**Condition 3.** *$F$ satisfies a local error-bound condition:*

$$\|F(z)\| \geq \tau \operatorname{dist}(z, Z^*)$$

*for all $z \in N(Z^*)$.*

**Condition 4.** *For $G \in \partial F(z)$ and any $\mu > 0$, $\|(G + \mu I)^{-1}\| \leq \frac{1}{\mu}$.*

**Remark 5.** *By our above analysis, if $F$ is monotone and Lipschitz continuous, Condition 4 is immediately satisfied.*

In summary, our convergence analysis establishes that the Newton steps of our hybrid algorithm, defined by Equation (5), achieves local quadratic convergence when applied to the residual operator $R_{\mathrm{DRS}}^{\gamma}(\cdot)$. Thus, our hybrid method effectively inherits the best-of-both worlds: global progress from FOMs and rapid local convergence of the regularized SSN method.

## 4 NUMERICAL RESULTS

We conduct numerical experiments on a variety of matrix games, including Kuhn poker (Kuhn, 1950), three families of matrix games arising from layered graph security settings (Černý et al., 2024), and random normal and uniform matrix games. In the realistic game instances, such as Kuhn poker and layered security games, $PRM^+$ rapidly achieves high-precision last-iterate convergence, making them less suitable for evaluating the benefits of our hybrid SSN algorithm. Accordingly, in this section, we focus on random normal and uniform matrix games, where $PRM^+$ struggles to efficiently converge to high-precision Nash equilibria.

Our hybrid algorithm is evaluated under two main schemes: **PSSN v1** and **PSSN v2**. These schemes perform a one-time switch from $PRM^+$ (QA) to the regularized semi-smooth Newton method once the duality gap of the $PRM^+$ iterates drops under a fixed threshold, but differ in their updating of the damping parameter; the SSN stage of PSSN v1 is initialized with a fixed damping parameter, whereas the SSN stage of PSSN v2 uses a damping parameter tuned by the $PRM^+$ steps. For completeness, we also include a third scheme, **HPSSN**, in the tables below. HPSSN follows a similar warm-starting strategy but alternates back and forth between $PRM^+$ and SSN at different rates based on the duality gap, in an effort to reduce the number of costly DRSSN steps while still exploiting curvature information near equilibrium. Although this adaptive alternation is conceptually appealing, in practice it is difficult to tune effectively: varying switching thresholds and the number of DRSSN steps does not consistently yield strong results without extensive tuning.

Our algorithms are compared to $PRM^+$ last-iterate (LI) and quadratic averaging (QA) baselines, which each use alternation, and are run for $5 \times 10^5$ iterations. All algorithms are initialized at $(x, y) = ((1/n)\mathbf{1}_n, (1/m)\mathbf{1}_m)$.

For succinctness, we report numerical results averaged over 10 independently generated random uniform and normal matrix games, of size $400 \times 800$. These games are representative of the general trends we observe across different random matrix game sizes and highlight the efficacy of our hybrid approach over FOMs for achieving high-precision solutions. In particular, despite the one-time cost of calculating resolvents to initialize our DRSSN method, our PSSN algorithms achieve significant speedups in the high-accuracy regime ($10^{-8}$ duality gap or lower), underscoring the practical benefit of incorporating curvature information. Note that, once DRSSN reaches a precision of around $10^{-8}$, it takes almost no additional time to reach higher levels of precision, e.g. $10^{-12}$. Thus, our numerical experiments corroborate the theoretical superlinear convergence guarantee of our methods.

All other numerical experiments, along with detailed descriptions of the algorithms and game instances, are provided in the appendix.

Table 1: Averaged clock times (in seconds) for $400 \times 800$ random uniform matrix games

| Method | Time to Reach Duality Gap Tolerance | | | | | |
| --- | --- | --- | --- | --- | --- | --- |
| | $10^{-2}$ | $10^{-4}$ | $10^{-6}$ | $10^{-8}$ | $10^{-10}$ | $10^{-12}$ |
| PCFR (LI) | 0.007 | 0.059 | 3.848 | 21.69 | | |
| PCFR (QA) | 0.006 | 0.030 | 0.293 | 6.424 | 21.97 | |
| PSSN v1 | 0.089 | 0.376 | 3.463 | 5.767 | 5.922 | 5.940 |
| PSSN v2 | 0.090 | 0.385 | 3.638 | 4.290 | 4.307 | 4.323 |
| HPSSN | 0.101 | 0.399 | 3.905 | 5.068 | 5.082 | 5.092 |

Table 2: Averaged clock times (in seconds) for $400 \times 800$ random normal matrix games

| Method | Time to Reach Duality Gap Tolerance | | | | | |
| --- | --- | --- | --- | --- | --- | --- |
| | $10^{-2}$ | $10^{-4}$ | $10^{-6}$ | $10^{-8}$ | $10^{-10}$ | $10^{-12}$ |
| PCFR (LI) | 0.006 | 0.087 | 3.325 | 23.51 | | |
| PCFR (QA) | 0.006 | 0.036 | 0.366 | 4.923 | 19.83 | |
| PSSN v1 | 0.128 | 0.690 | 4.238 | 5.267 | 5.289 | 5.306 |
| PSSN v2 | 0.128 | 0.691 | 3.983 | 5.055 | 5.077 | 5.094 |
| HPSSN | 0.125 | 0.572 | 3.895 | 4.786 | 4.790 | 4.815 |

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

## A  PROOFS

### A.1  PROOFS FOR SECTION 2

Recall that the DRS operator is given by

$$T_{\mathrm{DRS}}^{\gamma} = \mathrm{Id} - J_{N_{\mathcal{S}}}^{\gamma} + J_F^{\gamma} \circ (2J_{N_{\mathcal{S}}}^{\gamma} - \mathrm{Id}),$$

and the residual operator is given by $R_{\mathrm{DRS}}^{\gamma} = \mathrm{Id} - T_{\mathrm{DRS}}^{\gamma}$.

Because $F(\cdot)$ is linear and $N_{\mathcal{S}}(\cdot)$ is a normal cone operator, their resolvents simplify to

$$J_F^{\gamma}(z) = \begin{bmatrix} I & \gamma A \\ -\gamma A^{\top} & I \end{bmatrix}^{-1} z, \quad J_{N_{\mathcal{S}}}^{\gamma}(z) = \Pi_{\mathcal{S}}(z).$$

As a consequence, the DRS operator takes the form

$$T_{\mathrm{DRS}}^{\gamma} = \mathrm{Id} - \Pi_{\mathcal{S}} + \begin{bmatrix} I & \gamma A \\ -\gamma A^{\top} & I \end{bmatrix}^{-1} (2\Pi_{\mathcal{S}} - \mathrm{Id}).$$

*Proof of Lemma 1.* For any $z_1, z_2$, we have

$$
\begin{aligned}
& \|R_{\mathrm{DRS}}^{\gamma}(z_1) - R_{\mathrm{DRS}}^{\gamma}(z_2)\|^2 \\
={} & \|(z_1 - z_2) - (T_{\mathrm{DRS}}^{\gamma}(z_1) - T_{\mathrm{DRS}}^{\gamma}(z_2))\|^2 \\
={} & \|z_1 - z_2\|^2 + \|T_{\mathrm{DRS}}^{\gamma}(z_1) - T_{\mathrm{DRS}}^{\gamma}(z_2)\|^2 - 2(z_1 - z_2)^{\top}(T_{\mathrm{DRS}}^{\gamma}(z_1) - T_{\mathrm{DRS}}^{\gamma}(z_2)).
\end{aligned}
$$

It is known that $T_{\mathrm{DRS}}^{\gamma}$ is firmly nonexpansive (Ryu & Yin, 2022). Equivalently,

$$\|T_{\mathrm{DRS}}^{\gamma}(z_1) - T_{\mathrm{DRS}}^{\gamma}(z_2)\|^2 \le (z_1 - z_2)^{\top}(T_{\mathrm{DRS}}^{\gamma}(z_1) - T_{\mathrm{DRS}}^{\gamma}(z_2)).$$

Putting these pieces together yields

$$
\begin{aligned}
\|R_{\mathrm{DRS}}^{\gamma}(z_1) - R_{\mathrm{DRS}}^{\gamma}(z_2)\|^2 & \le \|z_1 - z_2\|^2 - (z_1 - z_2)^{\top}(T_{\mathrm{DRS}}^{\gamma}(z_1) - T_{\mathrm{DRS}}^{\gamma}(z_2)) \\
& = (z_1 - z_2)^{\top}(R_{\mathrm{DRS}}^{\gamma}(z_1) - R_{\mathrm{DRS}}^{\gamma}(z_2)).
\end{aligned}
$$

This implies that $R_{\mathrm{DRS}}^{\gamma}(\cdot)$ is monotone.

We also have

$$(z_1 - z_2)^{\top}(R_{\mathrm{DRS}}^{\gamma}(z_1) - R_{\mathrm{DRS}}^{\gamma}(z_2)) \le \|z_1 - z_2\| \|R_{\mathrm{DRS}}^{\gamma}(z_1) - R_{\mathrm{DRS}}^{\gamma}(z_2)\|.$$

Thus, we have

$$\|R_{\mathrm{DRS}}^{\gamma}(z_1) - R_{\mathrm{DRS}}^{\gamma}(z_2)\| \le \|z_1 - z_2\|.$$

This implies that $R_{\mathrm{DRS}}^{\gamma}(\cdot)$ is 1-Lipschitz. $\square$

*Proof of Lemma 2.* The strong semi-smoothness of $R_{\mathrm{DRS}}^{\gamma}(\cdot)$ follows from the existing results that establish the semi-smoothness of projection operators. Since the Euclidean projection onto the product of two simplices is a piecewise affine mapping (polyhedral projection), it is strongly semi-smooth (Qi & Sun, 1993). See also Facchinei & Pang (2003, Vol. II, Chapter 7) and Robinson (2009). As such, we have that $\Pi_{\mathcal{S}}(\cdot)$ is strongly semi-smooth. Since the strong semi-smoothness property is closed under scalar multiplication, summation, and composition, the residual map $R_{\mathrm{DRS}}^{\gamma}$ is strongly semi-smooth. $\square$

### A.2  PROOFS FOR SECTION 3

**Lemma 5** (Equivalence between Projection and Normal Cone). *Let $\mathcal{C}$ be a closed and convex set. Then $p \in \mathcal{C}$ is the projection of $x$ onto $\mathcal{C}$ if and only if $x - p \in N_{\mathcal{C}}(p)$.*

*Proof.* It is well-known that $p \in \mathcal{C}$ is the projection of $x$ onto $\mathcal{C}$ if and only if

$$\langle y - p, x - p \rangle \le 0$$

for all $y \in \mathcal{C}$. By the definition of the normal cone, this is equivalent to the condition that $x - p \in N_{\mathcal{C}}(p)$. $\square$

*Proof of Theorem 1 (i).* From Equation (3), a fixed point $z$ of $T_{\mathrm{DRS}}^{\gamma}$ satisfies

$$z = z - \Pi_{\mathcal{S}}(z) + (I + \gamma F)^{-1} (2\Pi_{\mathcal{S}}(z) - z)$$

or equivalently, $(I + \gamma F)^{-1} (2\Pi_{\mathcal{S}}(z) - z) = \Pi_{\mathcal{S}}(z)$. It follows that

$$(I + \gamma F)(\Pi_{\mathcal{S}}(z)) = 2\Pi_{\mathcal{S}}(z) - z.$$

Expanding and simplifying, we get that $\Pi_{\mathcal{S}}(z) + \gamma F(\Pi_{\mathcal{S}}(z)) = 2\Pi_{\mathcal{S}}(z) - z$, or equivalently,

$$\frac{1}{\gamma} (\Pi_{\mathcal{S}}(z) - z) = F(\Pi_{\mathcal{S}}(z)). \tag{9}$$

Furthermore, note that by the first-order optimality condition for projections, we have

$$\frac{1}{\gamma} \langle z - \Pi_{\mathcal{S}}(z), \tilde{z} - \Pi_{\mathcal{S}}(z) \rangle \leq 0$$

for all $\tilde{z} \in \mathcal{S}$, or equivalently,

$$\frac{1}{\gamma} (z - \Pi_{\mathcal{S}}(z)) \in N_{\mathcal{S}}(\Pi_{\mathcal{S}}(z)). \tag{10}$$

Summing Equation (9) and Equation (10), we get the desired result,

$$0 \in F(\Pi_{\mathcal{S}}(z)) + N_{\mathcal{S}}(\Pi_{\mathcal{S}}(z)).$$

By the optimality condition for a Nash equilibrium, $\hat{z} = \Pi_{\mathcal{S}}(z)$ is a Nash equilibrium. $\square$

*Proof of Theorem 1 (ii).* Since $\hat{z}$ is a Nash equilibrium, we have that $0 \in F(\hat{z}) + N_{\mathcal{S}}(\hat{z})$. Equivalently, $u := -F(\hat{z}) \in N_{\mathcal{S}}(\hat{z})$.

Consider $z := \hat{z} - \gamma F(\hat{z}) = \hat{z} + \gamma u$. Since $\hat{z} \in \mathcal{S}$ and $u \in N_{\mathcal{S}}(\hat{z})$ implies that $(\hat{z} + \gamma u) - \hat{z} = \gamma u \in N_{\mathcal{S}}(\hat{z})$, by Lemma 5, we have that $\hat{z} = \Pi_{\mathcal{S}}(\hat{z} + \gamma u)$. It follows that $\Pi_{\mathcal{S}}(z) = \Pi_{\mathcal{S}}(\hat{z} + \gamma u) = \hat{z}$, and we have that

$$\begin{aligned}
2\Pi_{\mathcal{S}}(z) - z &= 2\Pi_{\mathcal{S}}(\hat{z} + \gamma u) - (\hat{z} + \gamma u) \\
&= 2\hat{z} - (\hat{z} + \gamma u) \\
&= \hat{z} - \gamma u.
\end{aligned}$$

Applying $T_{\mathrm{DRS}}^{\gamma}$ to the lifted point $z$, we get that

$$\begin{aligned}
T_{\mathrm{DRS}}^{\gamma}(z) &= z - \Pi_{\mathcal{S}}(z) + (I + \gamma F)^{-1}(2\Pi_{\mathcal{S}}(z) - z) \\
&= (\hat{z} + \gamma u) - \hat{z} + (I + \gamma F)^{-1}(\hat{z} - \gamma u) \\
&= (\hat{z} + \gamma u) - \hat{z} + (I + \gamma F)^{-1}(\hat{z} + \gamma F(\hat{z})) \\
&= (\hat{z} + \gamma u) - \hat{z} + \hat{z} \\
&= \hat{z} + \gamma u = z
\end{aligned}$$

where the third line follows from the definition $u = -F(\hat{z})$. Thus, $z = \hat{z} - \gamma F(\hat{z})$ is a fixed point of $T_{\mathrm{DRS}}^{\gamma}$. $\square$

*Proof of Theorem 2.* For any $i \in \mathcal{A}$, the closed-form simplex projection formula gives

$$x_i = p_i - \alpha, \text{ where } \alpha = \frac{1}{|\mathcal{A}|} \left( \sum_{j \in \mathcal{A}} p_j - 1 \right)$$

For $i \notin \mathcal{A}$, we have $x_i = 0$. Differentiating with respect to $p_j$, we get, for $i \in \mathcal{A}$,

$$G_{ij} = \frac{\partial x_i}{\partial p_j} = \begin{cases} 1 - \frac{1}{|\mathcal{A}|}, & \text{if } i = j \in \mathcal{A} \\ -\frac{1}{|\mathcal{A}|} & \text{if } j \in \mathcal{A}, \ i \neq j \\ 0 & \text{if } j \notin \mathcal{A} \end{cases}$$

On the other hand, for $i \notin \mathcal{A}$, we trivially have $G_{ij} = 0$ for all $j$. Equivalently, the Jacobian admits the compact representation

$$G = \mathrm{diag}(a) - \frac{1}{\|a\|_1} aa^{\top}$$

where $a$ is the indicator vector of the active set $\mathcal{A}$. $\square$

**Lemma 6.** *For any two points $z_1 = (x_1 \, y_1) \in \mathcal{S}$ and $z_2 = (x_2 \, y_2) \in \mathcal{S}$, we have*

$$|\text{gap}(z_1) - \text{gap}(z_2)| \leq \sqrt{2}\|A\| \cdot \|z_1 - z_2\|.$$

*Proof of Lemma 6.* By definition,

$$\text{gap}(x, y) = \max_{\bar{y} \in Y} x^\top A \bar{y} - \min_{\bar{x} \in X} \bar{x}^\top A y.$$

Let us define $\Phi(x) := \max_{\bar{y} \in Y} x^\top A \bar{y}$ and $\Psi(y) := \min_{\bar{x} \in X} \bar{x}^\top A y$, so that $\text{gap}(x, y) = \Phi(x) + \Psi(y)$.

We have that

$$\begin{aligned}
\Phi(x_1) - \Phi(x_2) &= \max_{\bar{y} \in Y} x_1^\top A \bar{y} - \max_{\hat{y} \in Y} x_2^\top A \hat{y} \\
&\leq \max_{\bar{y} \in Y} \left[ x_1^\top A \bar{y} - x_2^\top A \bar{y} \right] \\
&= \max_{\bar{y} \in Y} \langle x_1 - x_2, A\bar{y} \rangle \\
&\leq \|A\| \cdot \|x_1 - x_2\|.
\end{aligned}$$

where the last line follows by the Cauchy-Schwarz inequality and the fact that $\|y\| \leq 1$ for all $y \in \Delta^n$. The above result also implies that

$$\Phi(x_2) - \Phi(x_1) \leq \|A\| \cdot \|x_2 - x_1\|.$$

Combining the above two results, we get that

$$|\Phi(x_1) - \Phi(x_2)| \leq \|A\| \cdot \|x_1 - x_2\|. \tag{11}$$

Similarly, we have that

$$\begin{aligned}
\Psi(y_1) - \Psi(y_2) &= \min_{\bar{x} \in X} \bar{x}^\top A y_1 - \min_{\hat{x} \in X} \hat{x}^\top A y_2 \\
&\leq \max_{\bar{x} \in X} \left[ \bar{x}^\top A y_1 - \bar{x}^\top A y_2 \right] \\
&= \max_{\bar{x} \in X} \langle \bar{x}, A(y_1 - y_2) \rangle \\
&\leq \|A\| \cdot \|y_1 - y_2\|
\end{aligned}$$

where the last line follows by the Cauchy-Schwarz inequality and the fact that $\|x\| \leq 1$ for all $x \in \Delta^m$. The above result also implies that

$$\Psi(y_2) - \Psi(y_1) \leq \|A\| \cdot \|y_2 - y_1\|.$$

Combining the above two results, we get that

$$|\Psi(y_1) - \Psi(y_2)| \leq \|A\| \cdot \|y_1 - y_2\|. \tag{12}$$

Since

$$\begin{aligned}
\text{gap}(z_1) - \text{gap}(z_2) &= \text{gap}(x_1, y_1) - \text{gap}(x_2, y_2) \\
&= (\Phi(x_1) - \Psi(y_1)) + (\Phi(x_2) - \Psi(y_2)) \\
&= (\Phi(x_1) - \Phi(x_2)) + (\Psi(y_1) - \Psi(y_2)),
\end{aligned}$$

we have that

$$\begin{aligned}
|\text{gap}(z_1) - \text{gap}(z_2)| &= |\Phi(x_1) - \Phi(x_2) + \Psi(y_1) - \Psi(y_2)| \\
&\leq |\Phi(x_1) - \Phi(x_2)| + |\Psi(y_1) - \Psi(y_2)| \\
&\leq \|A\| \cdot (\|x_1 - x_2\| + \|y_1 - y_2\|) \\
&\leq \sqrt{2}\|A\| \cdot \|z_1 - z_2\|
\end{aligned}$$

where the third line follows from applying the bounds of Equation (11) and Equation (12) and the final line follows from an application of the inequality $a + b \leq \sqrt{2}\sqrt{a^2 + b^2}$ where $a = \|x_1 - x_2\|$, $b = \|y_1 - y_2\|$, and so $\sqrt{a^2 + b^2} = \|z_1 - z_2\|$. $\qquad \square$

*Proof of Theorem 3 (i).* Let $z^*$ be the closest optimal solution to $z$ in $Z^*$. Rewriting the given condition $\|R_{\mathrm{DRS}}^\gamma(z)\| \geq \tau\|z - z^*\|$, we have

$$\|z - z^*\| \leq \frac{\|R_{\mathrm{DRS}}^\gamma(z)\|}{\tau}. \tag{13}$$

Let us define $\hat{z} := \Pi_{\mathcal{S}}(z)$ and $\hat{z}^* := \Pi_{\mathcal{S}}(z^*)$. Since projection is a nonexpansive operator, we have that

$$\|\hat{z} - \hat{z}^*\| \leq \|z - z^*\|. \tag{14}$$

Furthermore, by Lemma 6 applied to $\hat{z}$ and $\hat{z}^*$, we have that

$$|\mathrm{gap}(\hat{z}) - \mathrm{gap}(\hat{z}^*)| \leq \sqrt{2}\|A\| \cdot \|\hat{z} - \hat{z}^*\| \tag{15}$$

Combining the results of Equation (13), Equation (14), and Equation (15), we get that

$$|\mathrm{gap}(\hat{z}) - \mathrm{gap}(\hat{z}^*)| \leq \sqrt{2}\|A\| \cdot \|\hat{z} - \hat{z}^*\| \leq \sqrt{2}\|A\| \cdot \|\hat{z} - \hat{z}^*\| \leq \frac{\sqrt{2}\|A\| \cdot \|R_{\mathrm{DRS}}^\gamma(z)\|}{\tau}.$$

By Theorem 1 (i), $\mathrm{gap}(\hat{z}^*) = 0$. Thus, we conclude that

$$|\mathrm{gap}(\hat{z})| = |\mathrm{gap}(\Pi_{\mathcal{S}}(z))| \leq \frac{\sqrt{2}\|A\| \cdot \|R_{\mathrm{DRS}}^\gamma(z)\|}{\tau}. \qquad \square$$

*Proof of Theorem 3 (ii).* Let $\hat{z}^*$ be the closest Nash equilibrium to $\hat{z}$, so that $\mathrm{dist}(\hat{z}, \hat{Z}^*) = \|\hat{z} - \hat{z}^*\|$. By Equation (7), we have that

$$\|\hat{z} - \hat{z}^*\| \leq \frac{\mathrm{gap}(\hat{z})}{\delta(A)}. \tag{16}$$

Consider the lifted points $z := \hat{z} - \gamma F(\hat{z})$ and $z^* := \hat{z}^* - \gamma F(\hat{z}^*)$ of $\hat{z}$ and $\hat{z}^*$, respectively. Note that

$$\begin{aligned}
\|z - z^*\| &= \|(\hat{z} - \gamma F(\hat{z})) - (\hat{z}^* - \gamma F(\hat{z}^*))\| \\
&= \|\hat{z} - \hat{z}^* - \gamma(F(\hat{z}) - F(\hat{z}^*))\| \\
&\leq \|\hat{z} - \hat{z}^*\| + \gamma\|A\| \cdot \|\hat{z} - \hat{z}^*\| \\
&= (1 + \gamma\|A\|)\|\hat{z} - \hat{z}^*\|
\end{aligned} \tag{17}$$

where the third line follows from the triangle inequality and the definition of the operator $F$.

Finally, by Lemma 1 (ii), we know that $R_{\mathrm{DRS}}^\gamma(\cdot)$ is 1-Lipschitz, so

$$\|R_{\mathrm{DRS}}^\gamma(z)\| = \|R_{\mathrm{DRS}}^\gamma(z) - R_{\mathrm{DRS}}^\gamma(z^*)\| \leq \|z - z^*\|. \tag{18}$$

where the first equality follows by Theorem 1 (ii): $\hat{z}^*$ is a Nash equilibrium so its lift $z^*$ satisfies $R_{\mathrm{DRS}}^\gamma(z^*) = 0$.

Thus, we have that

$$\begin{aligned}
\|R_{\mathrm{DRS}}^\gamma(z)\| &\leq \|z - z^*\| \\
&\leq (1 + \gamma\|A\|)\|\hat{z} - \hat{z}^*\| \\
&\leq \frac{(1 + \gamma\|A\|)\mathrm{gap}(\hat{z})\|}{\delta(A)}
\end{aligned}$$

where the first line follows from Equation (18), the second line follows from Equation (17), and the final line follows from Equation (16). $\qquad \square$

*Proof of Theorem 4.* Let $z_k^*$ be the closest optimal solution (zero of $R_{\mathrm{DRS}}^\gamma$) to $z_k$. Note that by definition, for $G_k \in R_{\mathrm{DRS}}^\gamma(z_k)$, we have

$$\Delta z_k = -(G_k + \theta\|R_{\mathrm{DRS}}^\gamma(z_k)\|I)^{-1}R_{\mathrm{DRS}}^\gamma(z_k).$$

Adding and subtracting $z_k^* - z_k$ and factoring, we get that

$$\begin{aligned}
\Delta z_k &= -(G_k + \theta\|R_{\mathrm{DRS}}^\gamma(z_k)\|I)^{-1}R_{\mathrm{DRS}}^\gamma(z_k) \\
&= -(G_k + \theta\|R_{\mathrm{DRS}}^\gamma(z_k)\|I)^{-1}R_{\mathrm{DRS}}^\gamma(z_k) - (z_k^* - z_k) + (z_k^* - z_k) \\
&= -(G_k + \theta\|R_{\mathrm{DRS}}^\gamma(z_k)\|I)^{-1}[R_{\mathrm{DRS}}^\gamma(z_k) + (G_k + \theta\|R_{\mathrm{DRS}}^\gamma(z_k)\|I)(z_k^* - z_k)] + (z_k^* - z_k).
\end{aligned}$$

Taking the norm of both sides, it follows that

$$\|\Delta z_k\| = \left\|-(G_k + \theta\|R_{\mathrm{DRS}}^\gamma(z_k)\|I)^{-1}\left[R_{\mathrm{DRS}}^\gamma(z_k) + (G_k + \theta\|R_{\mathrm{DRS}}^\gamma(z_k)\|I)(z_k^* - z_k)\right] + (z_k^* - z_k)\right\|.$$

Applying the Triangle Inequality to the right-hand side and expanding gives

$$\begin{aligned}
\|\Delta z_k\| &= \left\|-(G_k + \theta\|R_{\mathrm{DRS}}^\gamma(z_k)\|I)^{-1}\left[R_{\mathrm{DRS}}^\gamma(z_k) + (G_k + \theta\|R_{\mathrm{DRS}}^\gamma(z_k)\|I)(z_k^* - z_k)\right] + (z_k^* - z_k)\right\| \\
&\leq \|(G_k + \theta\|R_{\mathrm{DRS}}^\gamma(z_k)\|I)^{-1}\| \cdot \|R_{\mathrm{DRS}}^\gamma(z_k) + G_k(z_k^* - z_k) + \theta\|R_{\mathrm{DRS}}^\gamma(z_k)\|(z_k^* - z_k)\| \\
&\quad + \|z_k^* - z_k\|.
\end{aligned}$$

Applying Lemma 3 on the first factor, we get that

$$\begin{aligned}
\|\Delta z_k\| &\leq \|(G_k + \theta\|R_{\mathrm{DRS}}^\gamma(z_k)\|I)^{-1}\| \cdot \|R_{\mathrm{DRS}}^\gamma(z_k) + G_k(z_k^* - z_k) + \theta\|R_{\mathrm{DRS}}^\gamma(z_k)\|(z_k^* - z_k)\| \\
&\quad + \|z_k^* - z_k\| \\
&\leq \frac{1}{\theta\|R_{\mathrm{DRS}}^\gamma(z_k)\|}\|R_{\mathrm{DRS}}^\gamma(z_k) + G_k(z_k^* - z_k) + \theta\|R_{\mathrm{DRS}}^\gamma(z_k)\|(z_k^* - z_k)\| + \|z_k^* - z_k\| \\
&\leq \frac{1}{\theta\|R_{\mathrm{DRS}}^\gamma(z_k)\|}\left(\|R_{\mathrm{DRS}}^\gamma(z_k) - G_k(z_k - z_k^*)\| + \theta\|R_{\mathrm{DRS}}^\gamma(z_k)\| \cdot \|z_k - z_k^*\|\right) + \|z_k^* - z_k\|.
\end{aligned}$$

where the final line follows from the Triangle Inequality. Since $R_{\mathrm{DRS}}^\gamma(\cdot)$ is strongly semi-smooth (Lemma 2), we know that

$$\|R_{\mathrm{DRS}}^\gamma(z_k) - G_k(z_k - z_k^*)\| \leq L_2\|z_k - z_k^*\|^2$$

for some $L_2 > 0$. Plugging this result in above, we get that

$$\begin{aligned}
\|\Delta z_k\| &\leq \frac{1}{\theta\|R_{\mathrm{DRS}}^\gamma(z_k)\|}\left(\|R_{\mathrm{DRS}}^\gamma(z_k) - G_k(z_k - z_k^*)\| + \theta\|R_{\mathrm{DRS}}^\gamma(z_k)\| \cdot \|z_k - z_k^*\|\right) + \|z_k^* - z_k\| \\
&\leq \frac{1}{\theta\|R_{\mathrm{DRS}}^\gamma(z_k)\|}\left(L_2\|z_k - z_k^*\|^2 + \theta\|R_{\mathrm{DRS}}^\gamma(z_k)\| \cdot \|z_k - z_k^*\|\right) + \|z_k^* - z_k\| \\
&= 2\|z_k - z_k^*\| + \frac{L_2}{\theta\|R_{\mathrm{DRS}}^\gamma(z_k)\|}\|z_k - z_k^*\|^2.
\end{aligned}$$

$$(19)$$

Since $R_{\mathrm{DRS}}^\gamma(\cdot)$ satisfies the local error-bound (Lemma 4), we have that

$$\|R_{\mathrm{DRS}}^\gamma(z_k)\| = \|R_{\mathrm{DRS}}^\gamma(z_k) - R_{\mathrm{DRS}}^\gamma(z_k^*)\| \geq \tau\|z_k - z_k^*\|.$$

Taking the reciprocal of both sides gives

$$\frac{1}{\|R_{\mathrm{DRS}}^\gamma(z_k)\|} \leq \frac{1}{\tau\|z_k - z_k^*\|}.$$

Plugging this bound into the result of Equation (19) gives

$$\begin{aligned}
\|\Delta z_k\| &\leq 2\|z_k - z_k^*\| + \frac{L_2}{\theta\|R_{\mathrm{DRS}}^\gamma(z_k)\|}\|z_k - z_k^*\|^2 \\
&\leq 2\|z_k - z_k^*\| + \frac{L_2}{\theta\tau\|z_k - z_k^*\|}\|z_k - z_k^*\|^2 \\
&= \left(2 + \frac{L_2}{\theta\tau}\right)\|z_k - z_k^*\|.
\end{aligned}$$

$$(20)$$

We now focus on the norm of the residual map at consecutive iterates $z_k$ and $z_{k+1}$. By definition,

$$\|R_{\mathrm{DRS}}^\gamma(z_{k+1})\| = \|R_{\mathrm{DRS}}^\gamma(z_k + \Delta z_k)\|.$$

Adding and subtracting $R_{\mathrm{DRS}}^\gamma(z_k) + G_k\Delta z_k$ and then applying the Triangle Inequality, we get that

$$\begin{aligned}
\|R_{\mathrm{DRS}}^\gamma(z_{k+1})\| &= \|R_{\mathrm{DRS}}^\gamma(z_k + \Delta z_k)\| \\
&= \|R_{\mathrm{DRS}}^\gamma(z_k + \Delta z_k) - R_{\mathrm{DRS}}^\gamma(z_k) - G_k\Delta z_k + R_{\mathrm{DRS}}^\gamma(z_k) + G_k\Delta z_k\| \qquad (21) \\
&\leq \|R_{\mathrm{DRS}}^\gamma(z_k + \Delta z_k) - R_{\mathrm{DRS}}^\gamma(z_k) - G_k\Delta z_k\| + \|R_{\mathrm{DRS}}^\gamma(z_k) + G_k\Delta z_k\|.
\end{aligned}$$

Since $R_{\mathrm{DRS}}^{\gamma}(\cdot)$ is strongly semi-smooth (Lemma 2), we once again have that

$$\|R_{\mathrm{DRS}}^{\gamma}(z_k + \Delta z_k) - R_{\mathrm{DRS}}^{\gamma}(z_k) - G_k \Delta z_k\| \le L_2 \|\Delta z_k\|^2. \tag{22}$$

On the other hand, by the definition of $\Delta z_k$, we have that

$$R_{\mathrm{DRS}}^{\gamma}(z_k) + (G_k + \theta \|R_{\mathrm{DRS}}^{\gamma}(z_k)\| I) \Delta z_k = 0.$$

Equivalently, after expanding, rearranging, and taking the norm of both sides, we have that

$$\|R_{\mathrm{DRS}}^{\gamma}(z_k) + G_k \Delta z_k\| = \theta \|R_{\mathrm{DRS}}^{\gamma}(z_k)\| \cdot \|\Delta z_k\|. \tag{23}$$

Plugging in the results of Equation (22) and Equation (23) into the final line of Equation (21), we get that

$$\begin{aligned}
\|R_{\mathrm{DRS}}^{\gamma}(z_{k+1})\| &\le \|R_{\mathrm{DRS}}^{\gamma}(z_k + \Delta z_k) - R_{\mathrm{DRS}}^{\gamma}(z_k) - G_k \Delta z_k\| + \|R_{\mathrm{DRS}}^{\gamma}(z_k) + G_k \Delta z_k\| \\
&\le L_2 \|\Delta z_k\|^2 + \theta \|R_{\mathrm{DRS}}^{\gamma}(z_k)\| \cdot \|\Delta z_k\|.
\end{aligned} \tag{24}$$

Substituting the result from Equation (20) into the result from Equation (24) and factoring, we get that

$$\begin{aligned}
\|R_{\mathrm{DRS}}^{\gamma}(z_{k+1})\| &\le L_2 \|\Delta z_k\|^2 + \theta \|R_{\mathrm{DRS}}^{\gamma}(z_k)\| \cdot \|\Delta z_k\| \\
&\le L_2 \left(2 + \frac{L_2}{\theta \tau}\right)^2 \|z_k - z_k^*\|^2 + \theta \|R_{\mathrm{DRS}}^{\gamma}(z_k)\| \left(2 + \frac{L_2}{\theta \tau}\right) \|z_k - z_k^*\| \\
&\le \left[L_2 \left(2 + \frac{L_2}{\theta \tau}\right)^2 + \theta L_1 \left(2 + \frac{L_2}{\theta \tau}\right)\right] \|z_k - z_k^*\|^2
\end{aligned} \tag{25}$$

where the final line follows as $R_{\mathrm{DRS}}^{\gamma}(\cdot)$ is 1-Lipschitz (Lemma 1 (ii)), or equivalently,

$$\|R_{\mathrm{DRS}}^{\gamma}(z_k)\| = \|R_{\mathrm{DRS}}^{\gamma}(z_k) - R_{\mathrm{DRS}}^{\gamma}(z_k^*)\| \le L_1 \|z_k - z_k^*\|.$$

Since $R_{\mathrm{DRS}}^{\gamma}(\cdot)$ satisfies the local error-bound condition (Lemma 4), we have that

$$\|R_{\mathrm{DRS}}^{\gamma}(z_k)\| = \|R_{\mathrm{DRS}}^{\gamma}(z_k) - R_{\mathrm{DRS}}^{\gamma}(z_k^*)\| \ge \tau \|z_k - z_k^*\|.$$

Squaring both sides and rearranging, it follows that

$$\|z_k - z_k^*\|^2 \le \frac{\|R_{\mathrm{DRS}}^{\gamma}(z_k)\|^2}{\tau^2}$$

Using the above result to bound the right-hand side of Equation (25) above, we get that

$$\begin{aligned}
\|R_{\mathrm{DRS}}^{\gamma}(z_{k+1})\| &\le \left[L_2 \left(2 + \frac{L_2}{\theta \tau}\right)^2 + \theta L_1 \left(2 + \frac{L_2}{\theta \tau}\right)\right] \|z_k - z_k^*\|^2 \\
&\le \frac{1}{\tau^2} \left[L_2 \left(2 + \frac{L_2}{\theta \tau}\right)^2 + \theta L_1 \left(2 + \frac{L_2}{\theta \tau}\right)\right] \|R_{\mathrm{DRS}}^{\gamma}(z_k)\|^2.
\end{aligned}$$

which gives us the desired local quadratic convergence of Equation (8). $\qquad \square$

## B ALGORITHM

### B.1 ADAPTIVE SCHEME

The $\beta_0$ coefficient serves as a contraction factor for the damping parameter $\lambda$, and its magnitude reflects the relative weight placed on second-order information from $G_k \in \partial R_{\mathrm{DRS}}^{\gamma}(x_k, y_k)$). As the iterates approach the optimal solution, the residual norm $\|R_{\mathrm{DRS}}^{\gamma}(x_k, y_k)\|$ decreases, in turn causing $\beta_0$ to shrink.

$\psi$ measures the quality of the Newton direction $\begin{bmatrix} \Delta x \\ \Delta y \end{bmatrix}$. A small value of $\psi$ indicates that the direction is not informative, while a large value indicates that the direction is informative and effectively

reduces the residual norm. Based on the value of $\psi$, the adaptive scheme, mirroring that of Lin et al. (2024), for updating $\lambda_{k+1}$ is as follows:

$$\lambda_{k+1} = \begin{cases} \max\{\underline{\lambda}, \beta_0 \lambda_k\} & \text{if } \psi \geq \alpha_2 \\ \beta_1 \lambda_k & \text{if } \alpha_1 \leq \psi < \alpha_2 \\ \min\{\overline{\lambda}, \beta_2 \lambda_k\} & \text{if } \psi < \alpha_1 \end{cases}$$

Note that this adaptive update rule distinguishes three regimes:

- **High-quality direction** ($\psi \geq \alpha_2$): the step produces significant progress toward an optimal solution, so the damping parameter is moderately decreased by multiplying by a contraction factor $\beta_0 < 1$.
- **Moderate-quality direction** ($\alpha_1 \leq \psi < \alpha_2$): the step produces moderate but not significant progress towards an optimal solution, so the damping parameter is moderately increased by a factor $\beta_1 > 1$.
- **Low-quality direction** ($\psi < \alpha_1$): the step produces little to no progress toward an optimal solution, so the damping parameter is significantly increased by a factor $\beta_2 \gg 1$.

In our experiments, we use the parameters $\alpha_1 = 10^{-2}$, $\alpha_2 = 5$, $\beta_1 = 2$, $\beta_2 = 5$, $\underline{\lambda} = 10^{-15}$, and $\overline{\lambda} = 10^{15}$.

### B.2 DESCRIPTION OF ALGORITHMS

**PSSN v1** performs a one-time switch from PRM$^+$ with quadratic averaging and alternation to the DRSSN method once the duality gap drops under some specified fixed threshold. After switching, the SSN method is initialized with a fixed damping parameter of 1 and is run exclusively thereafter. This makes PSSN v1 the simplest hybrid variant, but its rigidity in fixing the damping parameter can limit its speed across different problem instances.

**PSSN v2** also performs a one-time switch from PRM$^+$ with quadratic averaging and alternation to the DRSSN method once the duality gap drops under some specified fixed threshold. Every 500 PRM$^+$ iterations, the damping parameter is iteratively updated using the adaptive scheme. After switching, the SSN method is initialized with this updated damping parameter. As a result, PSSN v2 is often more adapted to the dynamics of the problem, and often achieves better stability and efficiency than PSSN v1, although at the cost of some additional updating during the PRM$^+$ phase.

**HPSSN** adopts a more dynamic approach, alternating between PRM$^+$ and DRSSN throughout the run. At different stages, the algorithm attempts to lift a PRM$^+$ iterate and switch into DRSSN, taking only a few Newton steps before deciding whether superlinear convergence is being achieved. If not, it projects the current iterate back onto $\mathcal{S}$ and switches back to PRM$^+$. This strategy attempts to reduce the number of system solves from the SSN steps while still exploiting curvature information near equilibrium, striking a balance between the low per-iteration cost of first-order methods and the fast local convergence of second-order methods. We reiterate the discussion of Section 4: though this idea is conceptually appealing, it is difficult to tune effectively in practice and depends heavily on each problem instance. Given these shortcomings, we only include it in the $400 \times 800$ matrix game experiments.

## C ADDITIONAL NUMERICAL EXPERIMENTS

We provide both detailed descriptions for each game instance and additional tests comparing our algorithms to the PRM$^+$ baselines.

### C.1 REALISTIC GAME INSTANCES

**Kuhn poker** (Kuhn, 1950). At the beginning of the game, the two players each pay one chip to the pot, and are dealt a single private card from a deck containing three cards: Jack, Queen, and King. The first player can check or bet, putting an additional chip in the pot. Then, the second player can check or bet after the first player's check, or fold/call the first player's bet. If the second player bets, the first player can decide whether to fold or to call the bet. At the showdown, the player with the

highest card who has not folded wins all the chips in the pot. The payoff matrix for Kuhn poker has dimension $27 \times 64$ with 690 nonzeros.

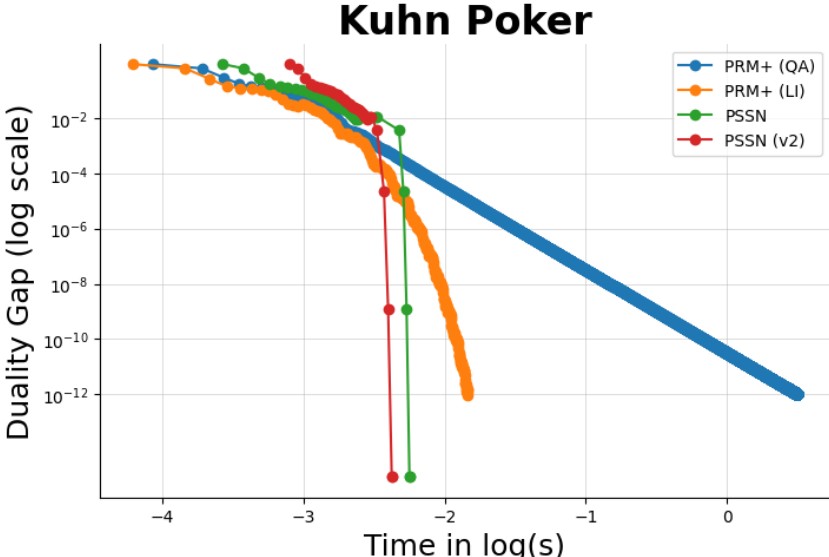

Kuhn poker is a standard benchmark in the game-theoretic literature and is widely used as a baseline for evaluating algorithms. However, its small size makes it straightforward to solve exactly. Even so, we observe improvements with our PSSN methods, which switch at a loose threshold of $10^{-2}$, over the last-iterate PRM$^+$ algorithm, and our methods also outperform PRM$^+$ with quadratic averaging and alternation.

**Layered graph security games** (Černý et al., 2024). We test on three classes of layered graph security games: the pursuit-evasion, logical-interdiction, and anti-terrorism games. Detailed descriptions of these instances can be found in Černý et al. (2024).

For all layered graph security game tests, we use the random seed 2025 together with the default parameter settings from the associated GitHub repository. The payoff matrix for the pursuit-evasion game has dimension $119 \times 887$ with 70253 nonzeros, the payoff matrix for the logical-interdiction game has dimension $119 \times 856$ with 2931 nonzeros, and the payoff matrix for the anti-terrorism game has dimension $119 \times 1080$ with 3756 nonzeros.

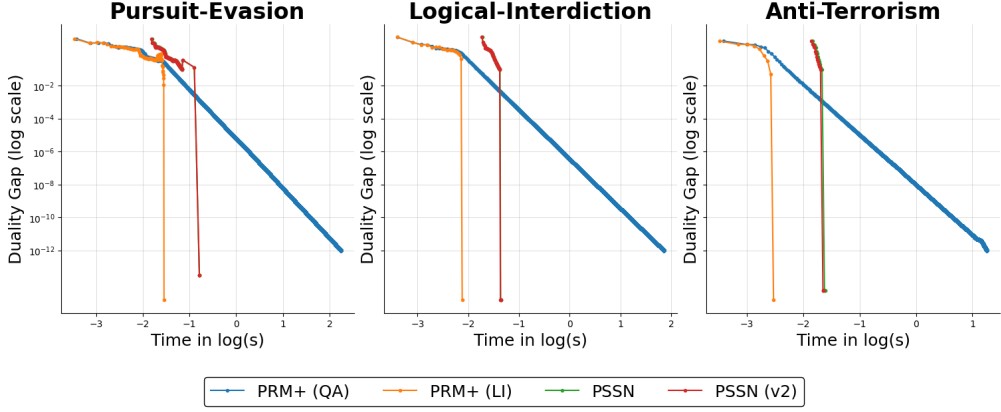

In these instances, the last-iterate PRM$^+$ algorithm solves the game directly and our PSSN methods solves the game quickly after switching at a relatively loose threshold of $10^{-1}$. We note that these instances clearly do not require the use of curvature information to find an efficient solution. However, our results still outperform PRM$^+$ with quadratic averaging and alternation, the de-facto approach to game solving. This motivates our focus on random matrix games in our experiments; in

random matrix instances, PRM$^+$ struggles to efficiently converge to a high-precision solution, and we find that our PSSN algorithms can do so extremely effectively.

## C.2 RANDOM GAME INSTANCES

The following tests are run on 10 independently generated random seeds across two classes of random matrices. In the uniform case, each entry $A_{ij}$ is drawn uniformly from $[-1, 1]$, while in the normal case, each entry $A_{ij}$ is drawn i.i.d from a standard normal distribution. We generate these instances on matrices of size $n \times m$ for $(n, m) = (100, 100), (400, 400)$, and $(400, 800)$.

We first test our methods across different duality gap switching thresholds. The best thresholds, in terms of lowest average clock times, are colored in blue.

Table 3: Averaged clock times (seconds) for $100 \times 100$ random matrix games vs thresholds.

| | *Random Uniform* | | | | | *Random Normal* | | | |
|---|---|---|---|---|---|---|---|---|---|
| **Method** | **Duality Gap Switching Threshold** | | | | **Method** | **Duality Gap Switching Threshold** | | | |
| | $10^{-1}$ | $10^{-2}$ | $10^{-3}$ | $10^{-4}$ | | $10^{-1}$ | $10^{-2}$ | $10^{-3}$ | $10^{-4}$ |
| PSSN v1 | 0.088 | 0.100 | 0.113 | 0.180 | PSSN v1 | 0.118 | 0.106 | 0.123 | 0.182 |
| PSSN v2 | 0.083 | 0.098 | 0.111 | 0.180 | PSSN v2 | 0.116 | 0.105 | 0.120 | 0.186 |

Table 4: Averaged clock times (seconds) for $400 \times 400$ random matrix games vs thresholds.

| | *Random Uniform* | | | | | *Random Normal* | | | |
|---|---|---|---|---|---|---|---|---|---|
| **Method** | **Duality Gap Switching Threshold** | | | | **Method** | **Duality Gap Switching Threshold** | | | |
| | $10^{-4}$ | $10^{-5}$ | $10^{-6}$ | $10^{-7}$ | | $10^{-4}$ | $10^{-5}$ | $10^{-6}$ | $10^{-7}$ |
| PSSN v1 | 3.154 | 2.645 | 4.213 | 6.866 | PSSN v1 | 3.762 | 2.645 | 4.519 | 7.132 |
| PSSN v2 | 2.939 | 2.424 | 17.269 | 23.693 | PSSN v2 | 3.446 | 3.068 | 26.445 | 39.878 |

Table 5: Averaged clock times (seconds) for $400 \times 800$ random matrix games vs thresholds.

| | *Random Uniform* | | | | | *Random Normal* | | | |
|---|---|---|---|---|---|---|---|---|---|
| **Method** | **Duality Gap Switching Threshold** | | | | **Method** | **Duality Gap Switching Threshold** | | | |
| | $10^{-4}$ | $10^{-5}$ | $10^{-6}$ | $10^{-7}$ | | $10^{-4}$ | $10^{-5}$ | $10^{-6}$ | $10^{-7}$ |
| PSSN v1 | 9.070 | 5.940 | 4.648 | 6.839 | PSSN v1 | 35.320 | 5.306 | 33.993 | 10.827 |
| PSSN v2 | 9.578 | 4.323 | 5.125 | 74.38 | PSSN v2 | 34.497 | 5.094 | 89.87 | 119.25 |

Across sizes and distributions, the best switching threshold is not universal and depends heavily on matrix dimension and problem structure. We note that smaller, better-conditioned matrix games (such as the randomly generated $100 \times 100$ instances) benefit from earlier or immediate switching, while larger, more ill-conditioned matrix games (such as the randomly generated $400 \times 400$ and $400 \times 800$ instances) require tighter thresholds for switching to the SSN phase. At the same time, we also note the tradeoff implicit in updates of the damping parameter: updating too aggressively can destabilize the SSN phase, whereas setting the threshold too tightly can delay the switch, causing PRM$^+$ to run excessively long before the benefits of second-order acceleration are realized.

Table 6: Averaged clock times (seconds) for $100 \times 100$ random matrix games.

*Random Normal*

| Method | **Time to Reach Duality Gap Tolerance** | | | | | |
|---|---|---|---|---|---|---|
| | $10^{-2}$ | $10^{-4}$ | $10^{-6}$ | $10^{-8}$ | $10^{-10}$ | $10^{-12}$ |
| PCFR (LI) | 0.008 | 0.120 | 4.476 | 31.205 | | |
| PCFR (QA) | 0.008 | 0.050 | 0.471 | 6.238 | 26.312 | |
| PSSN v1 | 0.035 | 0.105 | 0.116 | 0.117 | 0.118 | 0.118 |
| PSSN v2 | 0.035 | 0.102 | 0.114 | 0.115 | 0.115 | 0.116 |

*Random Uniform*

| Method | **Time to Reach Duality Gap Tolerance** | | | | | |
|---|---|---|---|---|---|---|
| | $10^{-2}$ | $10^{-4}$ | $10^{-6}$ | $10^{-8}$ | $10^{-10}$ | $10^{-12}$ |
| PCFR (LI) | 0.009 | 0.077 | 5.170 | 30.808 | | |
| PCFR (QA) | 0.008 | 0.041 | 0.396 | 8.452 | 27.946 | |
| PSSN v1 | 0.022 | 0.070 | 0.086 | 0.087 | 0.088 | 0.088 |
| PSSN v2 | 0.020 | 0.066 | 0.081 | 0.082 | 0.083 | 0.083 |

Table 7: Averaged clock times (seconds) for $400 \times 400$ random matrix games.

*Random Normal*

| Method | **Time to Reach Duality Gap Tolerance** | | | | | |
|---|---|---|---|---|---|---|
| | $10^{-2}$ | $10^{-4}$ | $10^{-6}$ | $10^{-8}$ | $10^{-10}$ | $10^{-12}$ |
| PCFR (LI) | 0.005 | 0.088 | 3.140 | 22.092 | | |
| PCFR (QA) | 0.005 | 0.034 | 0.332 | 4.258 | 18.042 | |
| PSSN v1 | 0.075 | 0.463 | 2.608 | 2.631 | 2.641 | 2.645 |
| PSSN v2 | 0.077 | 0.481 | 2.972 | 3.052 | 3.064 | 3.068 |

*Random Uniform*

| Method | **Time to Reach Duality Gap Tolerance** | | | | | |
|---|---|---|---|---|---|---|
| | $10^{-2}$ | $10^{-4}$ | $10^{-6}$ | $10^{-8}$ | $10^{-10}$ | $10^{-12}$ |
| PCFR (LI) | 0.006 | 0.054 | 3.608 | 20.130 | | |
| PCFR (QA) | 0.005 | 0.027 | 0.268 | 5.755 | 18.877 | |
| PSSN v1 | 0.075 | 0.375 | 2.538 | 2.629 | 2.640 | 2.645 |
| PSSN v2 | 0.075 | 0.375 | 2.319 | 2.410 | 2.417 | 2.424 |

In Table 6 and Table 7, we report average clock times across 10 independent random uniform and normal matrix games of sizes $100 \times 100$ and $400 \times 400$, respectively. For the smaller $100 \times 100$ instances, results are shown with a switching threshold of $10^{-1}$, which reflects the fact that such problems are well-conditioned and allow earlier switching into the DRSSN phase without loss of stability. In contrast, the results for the larger, more ill-conditioned $400 \times 400$ instances are shown with a tighter switching threshold of $10^{-5}$ to ensure that the DRSSN method is initialized within a local convergence regime. These thresholds correspond to the optimal switching values identified in Table 3 and Table 4, and are used here to provide an accurate performance comparison. Again, the results across these two tables highlight how the problem size and conditioning affect the choice of switching threshold, and illustrate the robustness of our hybrid method in both moderate and large-scale random matrix settings.

## D LLM USAGE

The authors used ChatGPT to aid/polish writing.

