# OpenReview forum: "A Direct Second-Order Method for Solving Two-Player Zero-Sum Games"
_ICLR.cc/2026/Conference — Submitted to ICLR 2026_

### Official Review · Reviewer_1xXe · 2025-10-30

**Soundness:** 3
**Presentation:** 3
**Contribution:** 3
**Rating:** 8
**Confidence:** 4

**Summary:**

The paper studies iterative methods for solving two-player zero-sum games, where the (approximate) Nash equilibrium is the solution concept. More specifically, the authors design a second order method that achieves quadratic convergence when it is sufficiently close to a Nash equilibrium. Moreover, they develop a lifting framework that allows the use of a first-order method— with Predictive Regret Matching+ chosen in the paper—to warm-start their algorithm. Experimentally, they observe an improved performance in a collection of games, using a few different strategies to switch between first and second order information.

**Strengths:**

The paper introduces a second order method which appears to be quite novel. Although some of the preliminary results are relatively straightforward to derive, there is a degree of technicality needed to obtain the main theorems. Finally, the experimental results also look promising. Overall, I would say that it is a well written paper with a clearly presented idea that is both mathematically sound and experimentally supported.

**Weaknesses:**

I think that the only weakness of the work is connected with the switch from first to second order information. That is in Theorem 4, there is no bound for $k$ (unless I missed something?) and the HPSSN method, which I agree with the authors on being the most interesting conceptually, is hard to tune. But the second point is ameliorated to a degree by the performance of PSSNs and could also be a topic of further work.

**Questions:**

1. As noted in the weakness part, is there any bound on the time needed to enter the local region?

2. Could you describe in greater detail the difficulty of finetuning HPSSN? (lines 447 - 450). Also, did you consider any other schemes?

I will also note here some typos and suggestions.

i. I did not immediately think of Bilinear Saddle point problem when I read the introduction title. You could consider using the full name, link to the problem statement or simply use "Introduction”.

ii. Given that the empty dot product can be used for a couple of operations in different products you may consider adding a sentence somewhere (we denote by $\circ$...)

iii. Similarly, you use $\hat Z ^\star$ in remark 3 and say “let $\hat Z ^*$ denote…” in the statement of Theorem 3. You could move it inside the remark.

iv. I think the statement of Theorem 4 should be altered a bit since you use both …$k$ sufficiently large… and $k\in N$. It should be $\{z_t\}, {t \ge k}$.

v. In line 744, there is a wrong sign.

vi. In line 769, there is a repetition of an expression.

---

> ### Author Response · Authors · 2025-11-17
>
> Thank you for taking the time to review our paper. We provide responses to your questions below.
>
> >  As noted in the weakness part, is there any bound on the time needed to enter the local region?
>
> We provide some details about an end-to-end convergence rate here, or a bound on the number of iterates required until a region of local quadratic convergence is reached after lifting. First, we require an additional instance-dependent parameter, $\epsilon(A, \gamma)$, in addition to the condition measure $\delta(A)$ of the payoff matrix (Definition 5) and its spectral norm $||A||$. The quantity $\epsilon(A, \gamma)$ characterizes the local region as measured by the norm of the operator $R_{\mathrm{DRS}}^{\gamma}$ at a point $z$: in particular, $||R_{\mathrm{DRS}}^{\gamma}(z)|| \leq \epsilon(A, \gamma)$ will guarantee that we are within the neighborhood in which quadratic convergence holds, as described in Theorem 4.
>
> Any end-to-end convergence rate will be dependent on these instance-dependent constants. We can show a theoretical guarantee that the local region is reached within $O(1/C)$ iterations, where
>
> $$ C = \frac{\epsilon(A, \gamma) \cdot \delta(A)}{\sqrt{2} ||A||}.$$
>
> This is guaranteed by Theorem 3(i): if the right-hand side of the inequality is at most $\epsilon(A, \gamma)$, then the lifted point has residual norm bounded by $\epsilon(A, \gamma)$. Consequently, the lifted iterate enters the region of local quadratic convergence precisely when the duality gap of the first-order method falls below $C$. Thus, a first-order method with an $O(1/T)$ duality gap rate, such as the Extragradient method, will in principle reach this regime after $O(1/C)$ iterations. (Technically, our rate under the Hybrid SSN method will be $O(1/\sqrt{C})$; however, our use of PRM+ despite its theoretical slower rate is due to its incredible practical performance.)
>
>
> We emphasize, however, that the Hybrid SSN method only realizes such an end-to-end rate when these instance-dependent parameters are known. Without direct knowledge of $\epsilon(A, \gamma)$ and $\delta(A)$, we cannot guarantee that a switch to the SSN phase places us in a region of local quadratic convergence. However, our experiments consistently indicate that this region is empirically quite large. In particular, we observe that the semi-smooth Newton method begins exhibiting fast local convergence even when the switch is made at a duality gap threshold of $10^{-5}$, or at a larger threshold depending on the specific class of matrix games.
>
>
>
> That being said, we would like to point out that the difficulty of working with instance-dependent parameters is not specific to our setting. Much of the literature relies on quantities such as the Hoffman constant (Hoffman 1952 [1]) for linear systems or condition measures (Mordukhovich et al. 2010 [2]) in game-theoretic and optimization settings, yet both are well known to be extremely difficult to compute or even approximate in practice. We view the development of understanding and estimating these constants as a theoretically important but challenging open direction, but one that extends beyond the focus of our present work.
>
> [1] Alan Hoffman. On Approximate Solutions of Systems of Linear Inequalities. Journal of Research of the National Bureau of Standards 1952.
>
> [2] Boris Mordukhovich, Javier Peña, and Vera Roshchina. Applying Metric Regularity to Compute a Condition Measure of a Smoothing Algorithm for Matrix Games. SIAM Journal on Optimization 2010.

---

> > ### Author Response · Authors · 2025-11-17
> >
> > > Could you describe in greater detail the difficulty of finetuning HPSSN? (lines 447 - 450). Also, did you consider any other schemes?
> >
> > Thanks for your question; we are happy to. The main difficulty in fine-tuning HPSSN lies in adapting and tuning the damping parameter used in Equation (5). We experimented with several schemes: varying the number of SSN steps taken upon switching based on the current residual norm or duality gap (e.g. taking more SSN steps when the duality gap is smaller), adjusting the damping parameter more aggressively when the iterates are far from optimality, and combinations of these heuristics and more. However, we found the damping parameter to be extremely sensitive, and small changes could dramatically affect stability and progress.
> >
> > A related issue is that frequent switching between the FOM and SSN stages can be counterproductive. Tuning the damping parameter too much during an SSN phase (especially when the iterates are not in the local region) can be unproductive and may counteract the tuning of the damping parameter done in the FOM. In contrast, if the SSN steps are not used to tune the damping parameter, then switching to the SSN method and taking SSN steps when not in the local region simply causes our algorithm to run longer without making great progress. Clearly, there is a tradeoff between taking additional SSN steps to tune the damping parameter and switching back to a FOM if we are not in a local region.
> >
> > In practice, we observed that once the iterate enters a good region, it is more effective to perform consecutive SSN steps to allow the damping parameter to stabilize, rather than alternating too frequently between the two methods. Thus, for both simplicity and effectiveness, our main algorithm involves switching only once from the FOM to the second-order phase. Nevertheless, our theory (Theorem 3) permits more frequent switching, and we believe that exploring such strategies further is a promising direction for future work in terms of improving practical performance.
> >
> >
> >
> > > I will also note here some typos and suggestions.
> >
> > Thank you for your suggestions and for reading our paper carefully! We will respond to each of them below.
> >
> > > i. I did not immediately think of Bilinear Saddle point problem when I read the introduction title. You could consider using the full name, link to the problem statement or simply use "Introduction”.
> >
> > We will update the section title to “Introduction to the Bilinear Saddle-Point Problem” in the final version to make the scope immediately clear.
> >
> > > ii. Given that the empty dot product can be used for a couple of operations in different products you may consider adding a sentence somewhere (we denote by …)
> >
> > We agree, and we will add a sentence when $\circ$ is first used.
> >
> > > iii. Notation of $\hat{Z}^*$ in Remark 3/Theorem 3
> >
> > We will introduce the relevant notation before presenting Remark 3 and Theorem 3 to ensure clarity.
> >
> > > iv. “I think the statement of Theorem 4 should be altered a bit …”
> >
> > We will adjust the statement accordingly to avoid confusion.
> >
> > > v. In line 744, there is a wrong sign.
> >    vi. In line 769, there is a repetition of an expression.
> >
> > We appreciate you catching these. Both issues have now been corrected.

---

> > > ### Author Response · Authors · 2025-11-24
> > >
> > > Thank you for your thoughtful feedback and your overall positive review of our paper. We hope our responses helped address your questions. If any additional questions come up or if further clarification would be helpful, we welcome further discussion.

---

### Official Review · Reviewer_Gh3G · 2025-10-31

**Soundness:** 3
**Presentation:** 2
**Contribution:** 2
**Rating:** 4
**Confidence:** 3

**Summary:**

This paper proposes a second-order method for computing Nash equilibria in two-player zero-sum games. The method is based on constructing an operator derived from the Douglas-Rachford splitting method. The fixed points of this operator in the lifted space correspond to Nash equilibria upon projection, and thus an equilibrium can be computed using a regularized semi-smooth Newton's method applied to the residual of the operator.  This second-order approach is used in conjunction with applying a first-order method (the paper particularly considers Predictive Regret Matching+) as a warm-start. The authors prove quadratic convergence of their second-order method in the local region near a Nash. Experimental results demonstrate the fast empirical convergence of this approach compared to using only a first-order method.

**Strengths:**

The (hybrid) second-order approach for computing NE in the two-player zero-sum game setting seems novel. The experimental results on random matrix games demonstrate the effectiveness of this method (compared to first-order methods), especially in the high-precision regime.

**Weaknesses:**

Overall, certain parts of the presentation are somewhat sloppy, which makes interpreting the paper's main theoretical contributions more challenging. In particular, the proposed second-order method relies on using a first-order method as a warm-start, but the main (local) convergence guarantee of the paper (Theorem 4) does not concretely connect the performance of the first-order method with the time to reach the local convergence regime. I believe the presentation would be strengthened if the authors were able to state a concrete end-to-end convergence rate for the Hybrid SSN method (Algorithm 1) using PRM+ for the warm start. It seems this could be achieved using the relationships between the residual norm and duality gap from Theorem 3. This would provide better theoretical insight into the empirical behavior of the algorithm observed in the experiments (where the main speedups occur in the high-precision regime). The authors also mainly suppose the first-order method is Predictive RM+, but it seems the framework could more generally use a different first-order method like Optimistic GDA.

Writing suggestions:
+ The presentation of Definition 5, Remark 3, and Theorem 3 needs improvement. For example, the definitions of $dist(\cdot,\cdot)$ and $\hat Z^*$ should be given before Definition 5.
+ The Hybrid SSN method framework (Algorithm 1) and the experimental results of the paper depend on the use of the PRM+ algorithm, however a description of PRM+ (other than one very high-level sentence in the introduction) is missing from the paper. Such a description should be included, at least in the appendix. See also the question below regarding whether PRM+ can be replaced with any other first-order method guaranteeing last-iterate convergence.

**Questions:**

Regarding the time to enter the local region in Theorem 4:
1. Using PRM+, can you provide a concrete quantitative bound on the number of iterates until the local quadratic convergence kicks in?
2. If instead of PRM+ you use some other first-order method with a (non-asymptotic) last-iterate convergence guarantee (e.g., Optimistic GDA). Again what can be said about the time until the fast local convergence, parameterized by the last-iterate convergence rate of the first-order method?
3. Also, can you clarify the qualifiers on $\tau$ in Theorem 4 (also in Lemma 4)? This is unclear as written.

---

> ### Author Response · Authors · 2025-11-17
>
> Thank you for taking the time to review our paper. We provide responses to your questions and concerns below:
>
> > The presentation of Definition 5, Remark 3, and Theorem 3 needs improvement. For example, the definitions of $dist(\cdot, \cdot)$ and $\hat{Z}^*$ should be given before Definition 5.
>
> Thanks for pointing this out. We will introduce the necessary notation before presenting these results.
>
> > The Hybrid SSN method framework (Algorithm 1) and the experimental results of the paper depend on the use of the PRM+ algorithm, however a description of PRM+ (other than one very high-level sentence in the introduction) is missing from the paper. Such a description should be included, at least in the appendix. See also the question below regarding whether PRM+ can be replaced with any other first-order method guaranteeing last-iterate convergence.
>
> We will include a formal description of PRM+ in the appendix for the final version, based on Gabriele Farina, Christian Kroer, and Tuomas Sandholm’s 2021 AAAI paper Faster Game Solving via Predictive Blackwell Approachability: Connecting Regret Matching and Mirror Descent.
>
> Thank you for your question about whether we can replace PRM+ with other FOMs. Indeed, our hybrid SSN framework is not tied to PRM+ specifically. The theoretical conditions (Theorem 3(ii)) governing the lift to our second-order semi-smooth Newton phase of our approach depend only on the quality of the iterate, as measured by the duality gap. Consequently, any first-order method that drives the duality gap down, including those guaranteeing last-iterate (or average) convergence in these zero-sum games, is compatible with our setup.
>
> However, our focus on PRM+ in our experiments is motivated by the fact that it is the state-of-the-art practical first-order method for large-scale zero-sum games and is widely used throughout the game-solving literature. Its strong empirical performance and scalability to EFGs make it a natural and representative choice for evaluating the hybrid approach.
>
> For completeness, we have included additional comparisons to EG and OGDA in Tables 1 and 2, which are presented below. In these experiments, we run both EG and OGDA for $10^5$ iterations; this choice is deliberate as each iteration is more expensive than a PRM+ iteration, and we can already clearly see the differences in both time and performance.
>
> *Table 1: Averaged clock times (in seconds) for 400 × 800 random normal matrix games*
>
> **Time to Reach Duality Gap Tolerance**
> | **Method**      | $10^{-2}$ | $10^{-4}$ | $10^{-5}$ | $9 \cdot 10^{-6}$ | $7.5 \cdot 10^{-6}$ | $10^{-6}$ |
> |-----------------|-----------|-----------|-----------|--------------------|-----------------------|-----------|
> | EG              | 0.322     | 7.255     | 46.532    | 51.580             | 59.049                | $(\ast)$  |
> | OGDA            | 0.070     | 1.584     | 10.378    | 11.381             | 13.342                | $(\ast)$  |
> | PRM$^{+}$ (LI)  | 0.006     | 0.087     | 0.465     | 0.505              | 0.592                 | 0.851     |
> | PRM$^{+}$ (QA)  | 0.006     | 0.036     | 0.110     | 0.116              | 0.131                 | 0.165     |
>
> ---
>
> *Table 2: Averaged clock times (in seconds) for 400 × 800 random uniform matrix games*
>
> **Time to Reach Duality Gap Tolerance**
> | **Method**      | $10^{-2}$ | $10^{-4}$ | $10^{-5}$ | $9 \cdot 10^{-6}$ | $10^{-6}$ |
> |-----------------|-----------|-----------|-----------|--------------------|-----------|
> | EG              | 0.436     | 9.645     | 65.302    | $(\dagger)$        | $(\ast)$  |
> | OGDA            | 0.099     | 2.140     | 14.361    | 16.340             | $(\ast)$  |
> | PRM$^{+}$ (LI)  | 0.007     | 0.059     | 0.301     | 0.344              | 3.848     |
> | PRM$^{+}$ (QA)  | 0.006     | 0.030     | 0.084     | 0.086              | 0.293     |
>
>
> As we can see, EG and OGDA exhibit significantly slower convergence than PRM+ across the full range of tolerances. Across the 10 random seeds we tested, the symbol $(\dagger)$ indicates that one seed failed to reach the duality gap threshold, and the symbol $(\ast)$ indicates that nine seeds failed to reach the duality gap threshold. In contrast, PRM+ attains such an iterate orders of magnitude more quickly. These results justify our use of PRM+ (with alternation under both quadratic averaging and last-iterate schemes) as both the primary baseline and the FOM used in our hybrid method.

---

> > ### Author Response · Authors · 2025-11-17
> >
> > > Using PRM+, can you provide a concrete quantitative bound on the number of iterates until the local quadratic convergence kicks in?
> > > If instead of PRM+ you use some other first-order method with a (non-asymptotic) last-iterate convergence guarantee (e.g., Optimistic GDA). Again what can be said about the time until the fast local convergence, parameterized by the last-iterate convergence rate of the first-order method?
> > > The presentation would be strengthened if the authors were able to state a concrete end-to-end convergence rate for the Hybrid SSN method (Algorithm 1) using PRM+ for the warm start. It seems this could be achieved using the relationships between the residual norm and duality gap from Theorem 3.
> >
> > Thank you for your questions and suggestions. We provide some details about an end-to-end convergence rate here, or a bound on the number of iterates required until a region of local quadratic convergence is reached after lifting, and answer the other questions along the way.
> >
> > First, we require an additional instance-dependent parameter, $\epsilon(A, \gamma)$, in addition to the condition measure $\delta(A)$ of the payoff matrix (Definition 5) and its spectral norm $||A||$. The quantity $\epsilon(A, \gamma)$ characterizes the size of the local region as measured by the norm of the operator $R_{\mathrm{DRS}}^{\gamma}$ at a point $z$: in particular, $||R_{\mathrm{DRS}}^{\gamma}(z)|| \leq \epsilon(A, \gamma)$ will guarantee that we are within the neighborhood in which quadratic convergence holds, as described in Theorem 4.
> >
> > Any end-to-end convergence rate will be dependent on these instance-dependent constants. In particular, we can show a theoretical guarantee that the local region is reached within $O(1/C)$ iterations, where
> >
> > $$ C  = \frac{\epsilon(A, \gamma) \cdot \delta(A)}{\sqrt{2} ||A||}.$$
> >
> > As you point out in Question 3, this can be guaranteed by Theorem 3(i): if the right-hand side of the inequality is at most $\epsilon(A, \gamma)$, then the lifted point has residual norm bounded by $\epsilon(A, \gamma)$. Consequently, the lifted iterate enters the region of local quadratic convergence precisely when the duality gap of the first-order method falls below $C$.
> > Thus, a first-order method with an $O(1/T)$ duality gap rate, such as the Extragradient method, will in principle reach this regime after $O(1/C)$ iterations. (Technically, our rate under the Hybrid SSN method will be $O(1/\sqrt{C})$; see our response below for why we focus on PRM+ despite its theoretical $O(1/\sqrt{T})$ duality gap rate.)
> >
> > We emphasize, however, that the Hybrid SSN method only realizes such an end-to-end rate when these instance-dependent parameters are known. Without direct knowledge of $\epsilon(A, \gamma)$ and $\delta(A)$, we cannot guarantee that a switch to the SSN phase places us in a region of local quadratic convergence. However, our experiments consistently indicate that this region is empirically quite large. In particular, we observe that the semi-smooth Newton method begins exhibiting fast local convergence even when the switch is made at a duality gap threshold of $10^{-5}$, or at a larger threshold depending on the specific class of matrix games.
> >
> > That being said, we would like to point out that the difficulty of working with instance-dependent parameters is not specific to our setting. Much of the literature relies on quantities such as the Hoffman constant (Hoffman 1952 [1]) for linear systems or condition measures (Mordukhovich et al. 2010 [2]) in game-theoretic and optimization settings, yet both are well known to be extremely difficult to compute or even approximate in practice. We view the development of understanding and estimating these constants as a theoretically important but challenging open direction, but one that extends beyond the focus of our present work.
> >
> > [1] Alan Hoffman. On Approximate Solutions of Systems of Linear Inequalities. Journal of Research of the National Bureau of Standards 1952.
> >
> > [2] Boris Mordukhovich, Javier Peña, and Vera Roshchina. Applying Metric Regularity to Compute a Condition Measure of a Smoothing Algorithm for Matrix Games. SIAM Journal on Optimization 2010.
> >
> > > Also, can you clarify the qualifiers on \tau in Theorem 4 (also in Lemma 4)? This is unclear as written.
> >
> > Thank you for pointing this out. In Theorem 4 (and Lemma 4), the qualifier on $\tau$ should simply read $\tau > 0$. While the metric subregularity constants may vary across different local neighborhoods associated with different optimal points, the $\tau$ appearing in the theorem can be understood as the infimum of these local values. In this sense, it serves as a single universal subregularity constant for our setting, analogous to how the Hoffman constant provides a uniform bound for linear systems.
> >
> > We hope this answers your question; we will include the qualifier and are certainly happy to include a comment about its interpretation in the final version.

---

> > > ### Author Response · Authors · 2025-11-24
> > >
> > > Thank you for your thoughtful review and for the points you raised. We hope our responses help clarify our contributions, improve our presentation, and address your concerns. If so, we would greatly appreciate a reconsideration of your score. We welcome further discussion and are happy to elaborate on any points that remain unclear.

---

### Official Review · Reviewer_7Fwu · 2025-10-31

**Soundness:** 3
**Presentation:** 3
**Contribution:** 2
**Rating:** 2
**Confidence:** 4

**Summary:**

The paper develops a direct second-order method for solving bilinear saddle-point problems. It shows that a semi-smooth Newton (SSN) method achieves local quadratic convergence. It then develops a more practical hybrid approach by leveraging predictive RM+ to warm start the SSN method. Experiments indicate that the proposed method is promising in the high-precision regime.

**Strengths:**

On the positive side, solving bilinear saddle-point problems is a central problem in game theory and optimization. Designing practical, scalable algorithms for solving such problems is an important and active research topic. The use of second-order Newton-type methods is relatively unexplored in this area. The paper contributes to filling this gap. While second-order methods have a significantly higher per-iteration complexity than first-order methods, such as PRM+, the paper proposes a natural hybrid approach whereby one only switches to the more expensive SSN method once a desired accuracy has been reached. Such hybrid approaches are very much prevalent in state of the art LP solvers, but they have been relatively unexplored in the context of solving zero-sum games. So pushing in that research direction is, I believe, worthwhile. Furthermore, the paper is very well written and organized. The key ideas are nicely exposed and sufficiently explained in the main body.

**Weaknesses:**

On the negative side, there are very basic flaws in the experimental evaluation of the paper, and the results are quite underwhelming. Concerning the results shown in Tables 1 and 2, the first basic issue is that there is no comparison with LP solvers. A commercial solver such as Gurobi would instantly solve exactly such small games. So on the whole, while the method is supposed to be superior in the high-precision regime, the paper fails to benchmark against the best approach in that regime, which is linear programming. In particular, it's hard to believe that Gurobi would be unable to solve a 400x800 game in less than 5 seconds. Besides failing to benchmark against LP solvers, the paper should certainly expand the first-order benchmarks considered. I should say that it is particularly strange to evaluate against the last-iterate of PRM+ when that's not guaranteed to even converge in zero-sum games. In practice, alternating PRM+ is typically much better than simultaneous PRM+, so at the very least the table should also contain results concerning alternating PRM+, together with other first-order methods that have linear convergence, such as extra-gradient and optimistic GD. The second main flaw in the experimental setup is that the games tested are very small, and it's hard to draw meaningful conclusions from random matrix games. There is a suite of benchmark extensive-form games that are commonly used for such purposes, but the paper doesn't provide those experiments, besides a quick mention about Kuhn poker where PRM+ finds exact solutions very rapidly. I would expect to see experiments on large extensive-form games to be convinced about the practical promise of the method. There are concerns that the proposed method doesn't scale well in larger games because of some of the preprocessing steps required.

I can only recommend acceptance for such paper if the experiments show promise. There are not many new insights in the theory of the paper. In particular, the whole hybrid approach is predicated on the assumption that the region of superlinear convergence will be entered reasonably soon; from a theoretical standpoint, that's not guaranteed. If the paper provides more comprehensive experiments that show the superiority of the method compared to state of the art solvers, I would certainly be in favor of acceptance.

**Questions:**

- Can the authors explain what the timeout in Tables 1 and 2 is? I wasn't able to find that.
- It is strange to use the term "PCFR" in Tables 1 and 2 since it is a normal-form game. I would suggest switching to PRM+ instead.

---

> ### Author Response · Authors · 2025-11-17
>
> Thank you for taking the time to review our paper. We provide responses to your concerns below:
>
> > Concerning the results shown in Tables 1 and 2, the first basic issue is that there is no comparison with LP solvers. A commercial solver such as Gurobi would instantly solve exactly such small games. So on the whole, while the method is supposed to be superior in the high-precision regime, the paper fails to benchmark against the best approach in that regime, which is linear programming. In particular, it's hard to believe that Gurobi would be unable to solve a 400x800 game in less than 5 seconds.
>
> We agree that LP-based formulations can provide an exact solution for small normal-form zero-sum games. However, our work is motivated by the computational regime relevant to modern game-solving, where first-order methods (FOMs) have become the standard due to their scalability and extremely low per-iteration cost. In particular, each iteration of these FOMs requires only matrix–vector multiplications and/or projections onto polyhedral sets, both of which are inexpensive and can leverage the sparsity present in many structured payoff matrices. In contrast, LP-based approaches (simplex, interior-point methods, etc.) require solving large linear systems at each iteration. These operations become prohibitively expensive at scales common in large-scale gaming solving.
>
> In this work, we develop a direct second-order method for game solving that can leverage the curvature information inherent in the underlying strategy spaces to obtain higher-precision solutions, something existing first-order methods struggle with and LP methods cannot handle at scale. As a first step toward this goal, we focus on the foundational matrix game setting.
>
> Thus, the intent of Tables 1 and 2 is not to compare against LP solvers on small instances, but rather to validate the theory and illustrate the hybrid method’s effectiveness compared to existing state-of-the-art FOMs in the high-precision regime for which it is designed. While we focus on matrix games in this paper, the theory we developed lays the groundwork for extending the approach to extensive-form games. See our responses below for further comments on the specific theoretical tools we introduce.
>
> > (Question 1) Can the authors explain what the timeout in Tables 1 and 2 is? I wasn't able to find that.
>
> In Tables 1 and 2, the timeouts correspond to a cap of $5 × 10^5$ iterations, as noted in line 453. We did not run any tests past this number of iterations because, for the methods that time out, the runtime already grows by several orders of magnitude before approaching the desired target accuracies. We will make this clearer in the final version of the paper.
>
> > (Question 2) It is strange to use the term "PCFR" in Tables 1 and 2 since it is a normal-form game. I would suggest switching to PRM+ instead.
>
> Thank you for the suggestion. We agree that PRM+ is the more appropriate term in the normal-form setting, and we have updated Tables 1 and 2 (and those in the Appendix) accordingly.

---

> > ### Author Response · Authors · 2025-11-17
> >
> > > I should say that it is particularly strange to evaluate against the last-iterate of PRM+ when that's not guaranteed to even converge in zero-sum games. In practice, alternating PRM+ is typically much better than simultaneous PRM+, so at the very least the table should also contain results concerning alternating PRM+, together with other first-order methods that have linear convergence, such as extra-gradient and optimistic GD.
> >
> > Thank you for the comment. We would like to clarify that our tables already include alternating PRM+. As noted in lines 452–453, our tables contain “PRM+ last-iterate (LI) and quadratic averaging (QA) baselines, which each use alternation,” and the PRM+ (QA) results are included in all tables. We agree that the simultaneous update rule is typically inferior, and that’s why we do all our benchmarking on the algorithms with alternation, which is the one used in practice.
> >
> > Regarding your suggestion to include extragradient (EG) or optimistic gradient descent/ascent (OGDA) baselines, we note that the game-solving literature overwhelmingly finds that PRM+/PCFR+ and its variants outperform other first-order methods on saddle-point problems. This is consistent with our own empirical evaluations. Nevertheless, for completeness, we are happy to report the relative EG and OGDA performances. For the 400x800 random matrix games tested in the main body, we provide the relative EG and OGDA results below. In these experiments, we run both EG and OGDA for $10^{5}$ iterations; this choice is deliberate as each iteration is more expensive than a PRM+ iteration, and we can already clearly see the differences in both time and performance.
> >
> >
> > *Table 1: Averaged clock times (in seconds) for 400 × 800 random normal matrix games*
> >
> > **Time to Reach Duality Gap Tolerance**
> > | **Method**      | $10^{-2}$ | $10^{-4}$ | $10^{-5}$ | $9 \cdot 10^{-6}$ | $7.5 \cdot 10^{-6}$ | $10^{-6}$ |
> > |-----------------|-----------|-----------|-----------|--------------------|-----------------------|-----------|
> > | EG              | 0.322     | 7.255     | 46.532    | 51.580             | 59.049                | $(\ast)$  |
> > | OGDA            | 0.070     | 1.584     | 10.378    | 11.381             | 13.342                | $(\ast)$  |
> > | PRM$^{+}$ (LI)  | 0.006     | 0.087     | 0.465     | 0.505              | 0.592                 | 0.851     |
> > | PRM$^{+}$ (QA)  | 0.006     | 0.036     | 0.110     | 0.116              | 0.131                 | 0.165     |
> >
> > ---
> >
> > *Table 2: Averaged clock times (in seconds) for 400 × 800 random uniform matrix games*
> >
> > **Time to Reach Duality Gap Tolerance**
> > | **Method**      | $10^{-2}$ | $10^{-4}$ | $10^{-5}$ | $9 \cdot 10^{-6}$ | $10^{-6}$ |
> > |-----------------|-----------|-----------|-----------|--------------------|-----------|
> > | EG              | 0.436     | 9.645     | 65.302    | $(\dagger)$        | $(\ast)$  |
> > | OGDA            | 0.099     | 2.140     | 14.361    | 16.340             | $(\ast)$  |
> > | PRM$^{+}$ (LI)  | 0.007     | 0.059     | 0.301     | 0.344              | 3.848     |
> > | PRM$^{+}$ (QA)  | 0.006     | 0.030     | 0.084     | 0.086              | 0.293     |
> >
> >
> > As we can see, EG and OGDA exhibit significantly slower convergence than PRM+ across the full range of tolerances. Across the 10 random seeds we tested, the symbol $(\dagger)$ indicates that one seed failed to reach the duality gap threshold, and the symbol $(\ast)$ indicates that nine seeds failed to reach the duality gap threshold. In contrast, PRM+ attains such an iterate orders of magnitude more quickly. These results justify our use of PRM+ (with alternation under both quadratic averaging and last-iterate schemes) as the primary baseline.
> >
> > > I would expect to see experiments on large extensive-form games to be convinced about the practical promise of the method. There are concerns that the proposed method doesn't scale well in larger games because of some of the preprocessing steps required.
> >
> > We appreciate this concern and agree that extending the method to large extensive-form games (EFGs) is an important direction for future work. In this work, we do not develop the theory for handling the treeplex projection needed in EFGs as an operator. However, we expect our overall framework to extend once the appropriate theory is developed.
> >
> > Here, our focus on matrix games helps us establish the foundational theory and demonstrate the practical viability of our hybrid method in the cleanest possible setting. As we highlight in our response to your next point, we have developed new theoretical tools even in this simpler regime, and we view extending them to large EFGs as a compelling and natural next step.

---

> > > ### Author Response · Authors · 2025-11-17
> > >
> > > > There are not many new insights in the theory of the paper. In particular, the whole hybrid approach is predicated on the assumption that the region of superlinear convergence will be entered reasonably soon; from a theoretical standpoint, that's not guaranteed.
> > >
> > > We respectfully disagree with the characterization that the paper lacks new theoretical insight. First and foremost, direct second-order methods have not previously been developed for game solving. Our work provides the first such algorithm, which achieves both strong numerical performance and enjoys superlinear theoretical convergence guarantees. In addition, we show how to effectively leverage progress made by any SOTA first-order method to warm-start our second-order SSN method.
> > >
> > > While the resulting implementation is not complicated (an advantage of the SSN method), the transition itself is highly nontrivial from a theoretical perspective. Even in standard convex optimization, orchestrating a principled transition from global first-order convergence to local superlinear Newton-type convergence is delicate, as the two methods operate on different operators, often in different ambient spaces, and rely on different regularity conditions. This difficulty is amplified in our setting, where the saddle-point operator of the direct game formulation is not monotone. As a result, we must instead work with the Douglas-Rachford splitting operator, which exploits the piecewise-polyhedral structure of the strategy simplex, and develop the SSN method and its analysis on this DRS formulation. Our theory formalizes the transition between the first-order and second-order regimes of our hybrid method, demonstrating both its theoretical soundness and practical effectiveness.
> > >
> > > In Theorem 3, we establish conditions under which the progress made by any first-order method that decreases the duality gap can be translated into progress with respect to the residual norm, ensuring compatibility with the SSN operator. In addition, in Theorem 2, we show that an element of the generalized Jacobian of our operator can be computed efficiently at each iteration, enabling the SSN method to be applied in practice. Finally, in Theorem 4, we prove that the resulting SSN iterations exhibit local quadratic convergence, providing a rigorous foundation for the second-order phase of our hybrid algorithm.

---

> > > > ### Author Response · Authors · 2025-11-24
> > > >
> > > > Thank you for your thoughtful review and for the points you raised. We hope our responses help clarify our contributions and address your concerns. If so, we would greatly appreciate a reconsideration of your score. We welcome further discussion and are happy to elaborate on any points that remain unclear.

---

> ### Comment · Reviewer_7Fwu · 2025-11-26
>
> I thank the reviewers for the reply.
>
> I maintain my original concern about the experimental evaluation of the paper. The experiments are very toy and do not make a convincing case in favor of the proposed method. I also don't think that excluding Gurobi makes for a fair comparison, although I read through the rationale given by the authors in their rebuttal. Nonetheless, as I said in my original review, I believe that the main approach of the paper is promising and original. If the authors expand the evaluation in a future version to larger games and show superiority over state of the art methods, the paper would make a solid contribution.

---

> > ### Author Response · Authors · 2025-12-02
> >
> > We thank you for your continued engagement and comments regarding the paper’s promise and originality.
> > We would like to respectfully argue that the current scope of the paper remains valid and scientifically significant.
> >
> > > Regarding LP comparisons for experiments
> >
> > In the modern game-solving literature, LP solvers are almost never used as experimental baselines. This applies to both experimental and theoretical works proposing new game-solving algorithms (whether CFR variants, optimistic methods, or recent approachability-based techniques). For instance, none of the five representative references below do so (and many more examples exist, including those mentioned in our response to reviewer RGok). To reiterate, this is because LP-based approaches are not competitive for realistic game sizes, and the field has converged on iterative first-order/regret-minimization methods as the practical standard.
> >
> > Regarding the comment that our experiments are on “toy games”, we note that many first-order methods are benchmarked precisely on the same random matrix games that we use (Gilpin et al. 2008 [1], Farina et al. 2019 [2], Fang et al. 2025 [4], Cai et al. 2025 [5]). In addition, we include experiments on several real-world instances arising from security games in the Appendix.
> >
> > Taken together, we believe our evaluation is appropriate, sufficient, and fully aligned with established community practices.
> >
> > > Regarding extension to EFGs
> >
> > Our contribution lies in developing the first direct second-order method for game solving, together with a rigorous theoretical framework that connects progress from any first-order method to the local superlinear convergence of our semi-smooth Newton (SSN) method. Extending this framework to large EFGs is a natural and important next step; however, we believe that establishing the theory in the matrix game setting is both necessary and sufficient, given its foundational nature and the novelty of our approach. Furthermore, we note that this progression from matrix games to EFGs follows the standard path in the game-solving literature, where new theoretical tools are typically first developed and validated in the simpler matrix setting before being extended to treeplex strategy spaces.
> >
> > We thank our reviewer again for the discussion; we appreciate it. For further discussion, we would like to point the reviewers and Area Chair to the high-level overview provided in the topmost comment.
> >
> > [1] Andrew Gilpin, Javier Peña, Tuomas Sandholm, First-Order Algorithm with $O(log⁡(1/\epsilon))$ Convergence for $\epsilon$-Equilibrium in Two-Person Zero-Sum Games, AAAI 2008.
> >
> > [2] Gabriele Farina, Christian Kroer, Tuomas Sandholm. Optimistic Regret Minimization for Extensive-Form Games via Dilated Distance-Generating Functions. NeurIPS 2019.
> >
> > [3] Darshan Chakrabarti, Julien Grand-Clément, Christian Kroer. Extensive-Form Game Solving via Blackwell Approachability on Treeplexes. NeurIPS 2024.
> >
> > [4] Zijian Fang, Zongkai Liu, Chao Yu, Chaohao Hu. Rapid Learning in Constrained Minimax Games with Negative Momentum. AAAI 2025.
> >
> > [5] Yang Cai, Gabriele Farina, Julien Grand-Clément, Christian Kroer, Chung-Wei Lee, Haipeng Luo, Weiqiang Zheng. Last-Iterate Convergence Properties of Regret-Matching Algorithms in Games. ICLR 2025.

---

### Official Review · Reviewer_RGok · 2025-10-31

**Soundness:** 2
**Presentation:** 3
**Contribution:** 2
**Rating:** 2
**Confidence:** 2

**Summary:**

This work proposes a direct solver for zero-sum matrix games. It is based on a Douglas-Rachford splitting operator of the residual operator into the sum of the residual of the unconstrained operator and an operator that encodes the constraint of both players strategies to lie inside the probability simplex. The resulting update is then computed using a semi-smooth Newton method.

The authors prove that the resulting method converges superlinearly and corroborate this claim with numerical examples on a series of numerical experiments.

**Strengths:**

The authors achieve their goal of designing a provably superlinear converging solver by what appears to be expert use of advanced techniques of non-smooth and convex analysis.

**Weaknesses:**

The first thing that comes to mind when reading the authors' claim of designing "the first direct second-order method for computing Nash equilibria in two-player zero-sum games" is that two-player matrix games are well-known to reduce to linear programming, for which a wide range of methods are already in existence.

The authors claim that "While this approach works in principle, it is impractical for large-scale games; even with state-of-the-art commercial solvers, the LP reformulation inflates the problem size and destroys exploitable structures in the payoff and constraint matrices, making exact solutions computationally prohibitive."

I have a hard time following this argument. After all, the reduction to linear programs yields problems of the form

$$
\max_{x, v} \quad  v \quad
\text{s.t.} \quad  A^\top x \ge v  \quad
\mathbf{1}_m^\top x = 1,
x \ge 0.
$$

It seems hard to agree that increasing the number of decision variables by 1 amounts to meaningfully "inflating the problem size." When it comes to the exploitation of problem-specific structure, no detail is provided as to how the proposed method would be able to exploit such structure in a way conventional LP solvers can't.

**Questions:**

As described under "weaknesses," I suspect the comparison to LP-based approaches to be misleading. If the authors can convince me otherwise, I would be more than happy to reconsider my recommendation.

---

> ### Author Response · Authors · 2025-11-17
>
> Thank you for taking the time to review our paper. We provide responses to your concerns below:
>
> > As described under "weaknesses," I suspect the comparison to LP-based approaches to be misleading. If the authors can convince me otherwise, I would be more than happy to reconsider my recommendation.
>
> We agree that LP-based formulations can provide an exact solution for normal-form zero-sum games. However, our comparison is motivated by the computational regime relevant to modern game-solving, where first-order methods (FOMs) have become the standard due to their scalability and extremely low per-iteration cost.
>
> In particular, each iteration of these FOMs requires only matrix–vector multiplications and/or projections onto polyhedral sets, both of which are inexpensive and can leverage the sparsity present in many structured payoff matrices. In contrast, LP-based approaches (simplex, interior-point methods, etc.) require solving large linear systems at each iteration. These operations become prohibitively expensive at scales common in large-scale gaming solving.
>
> This distinction is even more pronounced in extensive-form games  (EFGs), which generalize the matrix games we study. It is possible to write an LP whose size is linear in the size of the game tree. Even so, the high cost of the linear system solves involved in solving such an LP via black-box methods has led to LP approaches falling out of favor starting around 2007. These limitations have motivated the game-solving community to almost exclusively study iterative first-order regret-minimization algorithms, including but not limited to variants of CFR, CFR+ (Zinkevich et al. 2007 [2]; Tammelin 2014 [3]), and PCFR+ (Brown et al. 2019 [5]; Farina et al. 2021 [7]). Notably, the superhuman poker AIs Libratus (Brown & Sandholm 2017 [4]) and Pluribus (Brown & Sandholm 2019 [6]) are built entirely around such iterative first-order schemes, not LP-based solvers. As far as we know, the last paper that meaningfully used LP solving as part of a game solving pipeline for large EFGs was Gilpin and Sandholm (2007) [1] on solving Rhode-Island hold’em through abstraction followed by an LP solve.
>
> To summarize our work, we develop a direct second-order method for game solving that can leverage the curvature information inherent in the underlying strategy spaces to obtain higher-precision solutions, something existing first-order methods struggle with and LP methods cannot handle at scale. As a first step toward this objective, we focus on the more tractable matrix game setting and demonstrate both theoretically and empirically that progress made by scalable first-order methods (like PRM+, the state-of-the-art FOM for game-solving) can be effectively leveraged to warm-start a second-order semi-smooth Newton method, which can then be applied to obtain high-precision solutions efficiently.
>
> [1] Andrew Gilpin and Tuomas Sandholm. Lossless Abstraction of Imperfect Information Games. Journal of the ACM 2007.
>
> [2] Martin Zinkevich, Michael Johanson, Michael Bowling, and Carmelo Piccione. Regret Minimization in Games with Incomplete Information. NeurIPS 2007.
>
> [3] Oskari Tammelin. CFR+: Faster and More Efficient Counterfactual Regret Minimization. ArXiv 2014.
>
> [4] Noam Brown and Tuomas Sandholm. Libratus: The Superhuman AI for Heads-Up No-Limit Poker. Science 2017.
>
> [5] Noam Brown, Anton Lerer, Sam Gross, and Tuomas Sandholm. Deep Counterfactual Regret Minimization. ICLR 2019.
>
> [6] Noam Brown and Tuomas Sandholm. Superhuman AI for Multiplayer Poker. Science 2019.
>
> [7] Gabriele Farina, Christian Kroer, and Tuomas Sandholm. Faster game solving via predictive blackwell approachability: Connecting regret matching and mirror descent. AAAI 2021.

---

> > ### Author Response · Authors · 2025-11-24
> >
> > Thank you for your thoughtful review and for the points you raised. We hope our responses help clarify our contributions and address your concerns. If so, we would greatly appreciate a reconsideration of your score. We welcome further discussion and are happy to elaborate on any points that remain unclear.

---

### Author Response · Authors · 2025-12-02
**Summary of Reviewer Discussion**

Thank you to all the reviewers for their discussion and comments. Reviewers 7Fwu and 1xXe both note that the paper is well written, and reviewers 7Fwu, Gh3G, and 1xXe each highlight the novelty of our approach. Below, we summarize our discussion with the points raised by the reviewers.

> Regarding Comparison to LP-based approaches (and other first-order methods)

Reviewers RGok and 7Fwu raise questions about comparisons to LP-based solvers. As we emphasize in our responses, LP methods are not competitive in modern large-scale game-solving regimes, where the repeated solution of large linear systems is prohibitively expensive. Instead, first-order regret-minimization methods, such as Predictive Regret Matching+ (PRM+) and its variants, have long been the standard due to their scalability. It is true that for small games, LP methods would be fast. Even so, it has long been standard practice in the game-solving literature to benchmark on games that are small enough that we can easily run ablations and so on. Just to pick one example out of many, the Leduc game is used in almost every EFG-solving paper on first-order methods (Farina et al. 2019 [1], Farina et al. 2021 [2], Xu et al. 2024 [3], among many others). Leduc can be solved instantaneously by Gurobi! Even so, we do not make every paper include that fact in its numerics, because it is not pertinent to what we are trying to understand, which is the performance of methods that have the potential to scale beyond what is doable with black-box LP solvers.

We also emphasize that we did benchmark against the strongest variants of PRM+—quadratic averaging and last-iterate with alternation—as stated explicitly in the paper, though this was not acknowledged in reviewer 7Fwu’s follow-up. Furthermore, following questions by reviewers 7Fwu and Gh3G, we added additional experiments showing that first-order methods like OGDA and EG are substantially inferior to PRM+ in the game-solving setting. Finally, we emphasize that our lifting framework is not tied to PRM+: it is compatible with any first-order method that reduces the duality gap, allowing for a principled transition into the local region where the SSN method exhibits quadratic convergence. Our choice of PRM+ is substantiated by its strong numerical performance, as shown in our additional experiments.

> Regarding end-to-end convergence rate

Reviewers 7Fwu, Gh3G, and 1xXe each inquire about an end-to-end convergence rate for our hybrid algorithm (i.e., the time/iterations required to enter a local region after lifting). In our responses, we outline such a rate, which necessarily depends on several instance-dependent parameters.

We emphasize that these instance-dependent parameters (e.g., metric subregularity constants, condition measures) are well known to be difficult to compute or approximate. This limitation is not unique to our setting but reflects a general barrier in analyses of Newton-type methods (e.g., the Hoffman constant for solving linear systems).

Empirically, however, we find that the local region is quite large: across all our matrix game experiments, the SSN phase reliably exhibits fast local convergence even when switching at a duality gap threshold of $10^{-5}$ (or even earlier), which can be reached quickly through PRM+.

> Regarding Experiments on EFGs

Reviewer 7Fwu mentions that they would like to see experiments on extensive-form games (EFGs) to recommend acceptance. Our paper focuses on matrix games to develop the core theoretical framework. Given the substantial complexity of the matrix game analysis alone, we do not develop the treeplex projection and generalized Jacobian theory required for EFGs in this work. It is our opinion that it is another paper’s worth of work to extend our results to that setting. Nonetheless, we expect our overall framework to extend once the requisite theory is established, and we view extending our innovations to EFGs as a natural and compelling direction for future work.

We believe that we have adequately responded to all of the reviewers’ concerns. In summary, our work introduces the first direct second-order method for game solving, together with a rigorous framework for transitioning from global first-order progress to local superlinear convergence. This represents a conceptual and technical advancement beyond the existing game-solving literature, which has essentially exclusively studied first-order methods for nearly two decades.

[1] Gabriele Farina, Christian Kroer, Tuomas Sandholm. Optimistic Regret Minimization for Extensive-Form Games via Dilated Distance-Generating Functions. NeurIPS 2019.

[2] Gabriele Farina, Christian Kroer, and Tuomas Sandholm. Faster game solving via predictive blackwell approachability: Connecting regret matching and mirror descent. AAAI 2021.

[3] Haobo Xu, Zichang Fu, Jialin Xing. Minimizing Weighted Counterfactual Regret with Optimistic Blackwell Approachability. IJCAI 2024.

---

### Meta-Review · Area_Chair_etGb · 2025-12-25

**Summary:**

This work provides a variant of the Newton method to solve a two-player bilinear game. Some major concerns were raised by the reviewers that could not be fully addressed during the rebuttal phase; hence, this paper is recommended a reject. However, this should not discourage the authors. After incorporating the necessary updates as pointed out by the reviewers, this paper will be in good shape and is likely to make a solid contribution to the topic of two-player bilinear games.

Below is a list of major concerns that the next version of the paper may need to address:

- Reviewer 7Fwu has doubts about the experimental evaluation of this work. In particular, they think that the experiments on the linear systems are tiny and do not make a convincing case showing the significance of the proposed method. They also disagree with the authors' argument for excluding Gurobi in the comparison. They also think more benchmarks/datasets might be more appropriate. The reviewer explicitly stated the intention of remaining with the initial score after reviewing the authors' replies.

- Reviewer Gh3G raises the concern that the presentation is a little sloppy. They pointed out that the proposed second-order method relies on using a first-order method as a warm start, but the paper falls short of providing the time required to reach the local convergence regime described in Theorem 5. In response, the authors note that a slow sublinear rate is required to enter this regime when using some results. However, the reply might raise doubts about the empirical comparison with existing LP solvers, as the proposed method needs a warm-start. This issue is also relevant to the comment of Reviewer 7Fwu regarding the lack of a comparison of some LP solvers. For this, the authors in the rebuttal reiterated that LP methods are not competitive in modern large-scale game-solving regimes. But the experiment settings in this paper are not large-scaled as also acknowledged by the authors; hence, the comparisons in the experiments should include these LP solvers so that the significance of the proposed method can be made clearer.

In addition to updating the paper based on the above comments, the condition of local convergence may need to be made more explicit to make the theoretical results more transparent (i.e., when to have the super-linear convergence).

**Reviewer Concerns:**

The list of major outstanding concerns that the next version of the paper may need to address can be found on the *Summary* section above.

There is a request for more empirical evidence. The authors did make effort in providing new experimental results, but unfortunately, the results are not fully appreciated by the reviewers.

**Reviewer Scores:**

- The concerns of Reviewers RGok and 7Fwu regarding the empirical evidence remain, and it is therefore unlikely that they will change their scores.

- Reviewer Gh3G has a concern about the presentation of the local convergence guarantee. Specifically, since the radius/condition of local convergence is not explicit in Theorem 4, the concern still stands.

---

### Decision · Program_Chairs · 2026-01-26

Reject